# The RIF1-long splice variant promotes G1 phase 53BP1 nuclear bodies to protect against replication stress

Lotte P Watts[1], Toyoaki Natsume[2,3], Yuichiro Saito[2], Javier Garzon[1], Qianqian Dong[1], Lora Boteva[4], Nick Gilbert[4], Masato T Kanemaki[2,3], Shin-ichiro Hiraga[1], Anne D Donaldson[1]*

[1]Institute of Medical Sciences, University of Aberdeen, Aberdeen, United Kingdom; [2]Department of Chromosome Science, National Institute of Genetics, Research Organization of Information and Systems (ROIS), Mishima, Japan; [3]Department of Genetics, The Graduate University for Advanced Studies (SOKENDAI), Mishima, Japan; [4]MRC Human Genetics Unit, The University of Edinburgh, Edinburgh, United Kingdom

**Abstract** Human cells lacking RIF1 are highly sensitive to replication inhibitors, but the reasons for this sensitivity have been enigmatic. Here, we show that RIF1 must be present both during replication stress and in the ensuing recovery period to promote cell survival. Of two isoforms produced by alternative splicing, we find that RIF1-Long alone can protect cells against replication inhibition, but RIF1-Short is incapable of mediating protection. Consistent with this isoform-specific role, RIF1-Long is required to promote the formation of the 53BP1 nuclear bodies that protect unrepaired damage sites in the G1 phase following replication stress. Overall, our observations show that RIF1 is needed at several cell cycle stages after replication insult, with the RIF1-Long isoform playing a specific role during the ensuing G1 phase in damage site protection.

*For correspondence:
a.d.donaldson@abdn.ac.uk

Competing interests: The authors declare that no competing interests exist.

## Introduction

The RIF1 protein has emerged as a central regulator of chromosome maintenance, acting in double-strand break repair and DNA replication control (*Chapman et al., 2013*; *Escribano-Díaz et al., 2013*; *Hiraga et al., 2017*). During double-strand break repair, RIF1 is recruited by 53BP1, dependent upon phosphorylation of 53BP1 by ATM (*Chapman et al., 2013*; *Escribano-Díaz et al., 2013*; *Di Virgilio et al., 2013*). Together RIF1 and 53BP1 recruit Shieldin and suppress BRCA1 recruitment to damage sites, opposing homologous recombination-based repair and favouring non-homologous end joining (*Bunting et al., 2010*; *Setiaputra and Durocher, 2019*).

RIF1 is also implicated in protecting cells from replication stress (*Buonomo et al., 2009*; *Kumar and Cheok, 2014*; *Mazouzi et al., 2016*). Replication stress can be induced by various conditions including drugs such as Aphidicolin, which interrupts replication fork progression by inhibiting the replicative DNA polymerases α, δ and ε (*Syvaoja et al., 1990*). Replication stress leads to genomic instability, mutation and eventually disease (*Burrell et al., 2013*; *Kerzendorfer et al., 2013*; *Ogi et al., 2012*), so understanding the cellular response is central for understanding accurate genome duplication and the action of replication inhibitors as anti-cancer drugs (*Feng et al., 2003*; *Imai et al., 2016*). RIF1-deficient cells are acutely sensitive to replication stress, in fact appearing to be more sensitive to replication inhibitors than to double-strand break-inducing agents (*Buonomo et al., 2009*), suggesting that protection from stress is a critical RIF1 function.

Several roles have been described for RIF1 in replication control. RIF1 acts as a Protein Phosphatase 1 (PP1) 'substrate-targeting subunit' (*Peti et al., 2013*) that suppresses replication origin

initiation by directing PP1 to dephosphorylate the MCM replicative helicase complex (*Alver et al., 2017*; *Davé et al., 2014*; *Gnan et al., 2020*; *Hiraga et al., 2017*; *Hiraga et al., 2014*; *Mattarocci et al., 2014*). RIF1 moreover stimulates origin licensing during G1 phase, and protects replication forks from unscheduled degradation (*Garzón et al., 2019*; *Hiraga et al., 2017*; *Mukherjee et al., 2019*). However, whether deficiency in these functions accounts for the replication stress sensitivity of cells lacking RIF1 has remained unclear (*Buonomo et al., 2009*; *Feng et al., 2003*).

RIF1 also acts in mitosis to maintain genomic stability. During anaphase, RIF1 is recruited to ultra-fine anaphase bridges (UFBs), along with the BLM and PICH proteins that ensure proper chromosome segregation (*Hengeveld et al., 2015*). UFBs are believed to correspond to unresolved topological entanglements that escape checkpoint surveillance and persist into mitosis (*Bergoglio et al., 2013*; *Bhowmick et al., 2016*). Unresolved DNA damage that passes to daughter cells leads to the formation of 53BP1 nuclear bodies during G1 phase, which protect the damaged DNA (*Bruhn et al., 2014*; *Lukas et al., 2011*; *Moreno et al., 2016*). RIF1 has also recently been described as functioning at the midbody during cytokinesis (*Bhowmick et al., 2019*).

The human RIF1 transcript undergoes alternative splicing producing two protein isoforms: a long variant of 2472 amino acids ('RIF1-L'), and a short variant ('RIF1-S') which lacks 26 amino acids close to the C-terminus of the protein (*Xu and Blackburn, 2004*). RIF1-S was reported to be more abundant in various cancer cell lines (*Xu and Blackburn, 2004*), hinting at distinct effects of the isoforms. Although RIF1-L was designated the canonical form, studies using cloned RIF1 have predominantly used RIF1-S (*Batenburg et al., 2017*; *Escribano-Díaz et al., 2013*; *Xu et al., 2010*; *Xu and Blackburn, 2004*), without testing for distinct functions of the isoforms.

## Results

### Analysing fluorescent degron-tagged RIF1 reveals highly dynamic cell cycle localisation

We aimed to understand how RIF1 guards against replication stress. First, we confirmed in a colony formation assay (CFA) that HEK293 cells depleted for RIF1 are sensitive to the DNA polymerase inhibitor Aphidicolin (*Figure 1A,B*). HCT116 cells deleted for both copies of *RIF1* were also sensitive (*Figure 1C*). Doses of Aphidicolin were designed to slow replication fork progression and induce replication stress, as opposed to blocking the cell cycle (*Buonomo et al., 2009*). These results imply a specific role for RIF1 in protecting cells under replication stress conditions.

Since RIF1 functions at various cell cycle stages, we explored when RIF1 is needed to maintain cell proliferation following replication stress. Specifically, we tested if RIF1 function is required during DNA replication stress, after its occurrence, or both during and after stress. Using auxin-inducible degron (AID) technology we constructed a cell line allowing rapid depletion and re-expression of RIF1 at different phases of the cell cycle (*Natsume et al., 2016*; *Nishimura et al., 2009*). In an HCT116-based cell line carrying the auxin-responsive degron recognition protein OsTIR1 under doxycycline (DOX) control, we N-terminally tagged both RIF1 genomic copies with a degron-Clover construct, termed 'mAC', consisting of a <u>m</u>ini-<u>a</u>uxin-inducible degron and monomer <u>C</u>lover (a derivative of GFP) (*Figure 1D*; *Yesbolatova et al., 2019*). The expressed construct remains under control of the endogenous RIF1 promoter. Western blot analysis indicated that expression levels of mAC-RIF1 in the absence of Auxin were similar to those of endogenous untagged RIF1 (*Figure 1E*). Treatment for 24 hr with DOX and Auxin led to near-complete degradation of RIF1, as visualised by western blotting (*Figure 1E*) and microscopy (*Figure 1F*). Using flow cytometry analysis of the RIF1-fused mClover tag, we established suitable concentrations of DOX and Auxin to allow effective depletion (*Figure 1—figure supplement 1A,B*). Degradation was largely complete after 3 hr of Auxin treatment (*Figure 2—figure supplement 1A*), whilst expression was restored to almost normal levels 5 hr after Auxin removal (*Figure 2—figure supplement 1B*). Together, these results confirm the construction of a cell line allowing rapid depletion and re-expression of RIF1.

We visualised mAC-RIF1 based on Clover fluorescence, in the first study of the dynamic behaviour of tagged endogenous RIF1 in living cells. In live-cell imaging experiments we observed a pattern of numerous mAC-RIF1 foci throughout S phase nuclei (*Figure 1G*, top row), often with 3–6 prominent foci superimposed on a pattern of more numerous smaller foci, consistent with previous

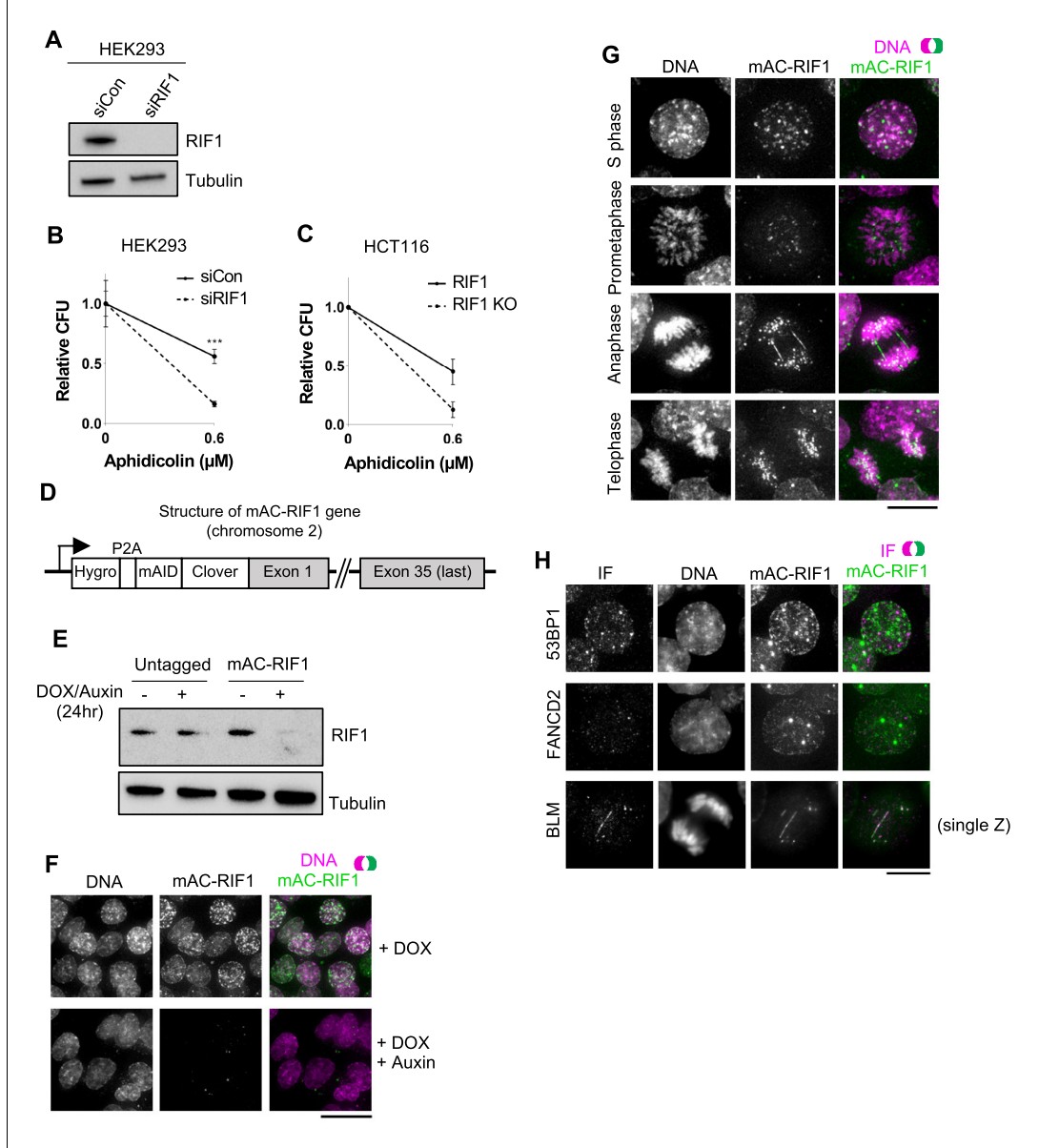

**Figure 1.** Characterisation of HCT116-based cell lines with auxin-inducible Degron-tagged RIF1. (**A**) Confirmation of siRIF1 efficacy three days after siRNA transfection. Whole cell protein extracts were analysed by western blotting with anti-RIF1 antibody. Tubulin shown as a loading control. (**B**) Colony Formation Assay (CFA) confirming Aphidicolin sensitivity of HEK293 cells treated with siRIF1. Plot shows mean and range of technical triplicates. \*\*\*p<0.001. (**C**) CFA testing Aphidicolin sensitivity of HCT116 RIF1-KO cells. 'RIF1' cell line is HCT116 mAC-RIF1. Plot shows mean and range from two biological replicates (each performed in technical triplicate). (**D**) Structure of auxin-inducible degron (AID)-tagged RIF1 construct (mAC-RIF1), located at both endogenous RIF1 loci in HCT116 cells carrying the auxin-responsive F-box protein *Oryza sativa* TIR1 (OsTIR1) under DOX control. The RIF1 gene is fused to a tag containing a self-cleaving P2A peptide, hygromycin resistance marker, mini-auxin-inducible degron (mAID) and monomer Clover (mClover) protein. After self-cleavage at the P2A, RIF1 protein is expressed as N-terminal in-frame fusion with mAID and Clover. (**E**) Confirmation of mAC-RIF1 protein degradation. Cells were incubated with 2 μg/ml DOX and 500 μM Auxin for 24 hr, then protein extracts analysed by western blotting with antibody against RIF1. Tubulin is shown as a loading control. Testing of drug concentrations is shown in *Figure 1—figure supplement 1*. (**F**) mAC-RIF1 degradation assessed by microscopy. mAC-RIF1 cells were treated with 2 μg/ml DOX and 500 μM Auxin for 24 hr. DNA was stained with SiR-DNA (magenta). Scale bar = 10 μm. (**G**) Examples of mAC-RIF1 localisation at different cell cycle stages. DNA stained with SiR-DNA. Scale bar = 10 μm. (**H**) mAC-RIF1 co-localises with BLM at UFBs but not with 53BP1 or FANCD2 repair proteins. Fixed cells were stained with the above-mentioned antibodies. Scale bar = 10 μm.

The online version of this article includes the following figure supplement(s) for figure 1:

**Figure supplement 1.** Characterisation and testing of mAC-RIF1 depletion.

immunofluorescence studies (*Buonomo et al., 2009*; *Xu and Blackburn, 2004*; *Yamazaki et al., 2012*). In prometaphase mAC-RIF1 exhibits a novel, circumpolar localisation pattern similar to that described for kinetochores (*Figure 1G*, second row, showing polar view of condensed chromosomes) (*Magidson et al., 2015*). At anaphase, even in unperturbed cells, mAC-RIF1 was sometimes observed at bridge structures (*Figure 1G*, third row) where it colocalised with the bridge marker protein BLM, but not always with conventional DNA dyes (*Figure 1H*, bottom panel) (*Barefield and Karlseder, 2012*; *Chan et al., 2007*; *Hengeveld et al., 2015*). At telophase, mAC-RIF1 formed multiple small foci associated with separated chromosomes (*Figure 1G*, bottom row), as described previously (*Xu and Blackburn, 2004*; *Yamazaki et al., 2012*). Time-lapse imaging of HCT116 mAC-RIF1 cells containing an mCherry-tagged PCNA revealed intense RIF1 foci that accumulated through S phase and G2 (*Video 1*), but disappeared by metaphase (*Video 2*). RIF1 signal was absent for a short period at metaphase, quickly followed by reappearance of numerous smaller foci on telophase chromosomes.

The mAC-RIF1 fusion therefore retains the localisation and functional characteristics of the endogenous RIF1 protein. Its highly dynamic behaviour as revealed by this live-cell study indicates that RIF1 functions in several cell cycle phases to maintain chromosome stability.

## RIF1 is needed during and after replication stress to promote cell proliferation

Prolonged auxin-induced degradation of RIF1 caused sensitivity to Aphidicolin, comparable to an HCT116 RIF1-knockout cell line (RIF1 KO) (*Figure 2A*, *Figure 2—figure supplement 2A*, *Figure 3B* lane 4). To investigate when RIF1 function is required to protect against the effects of Aphidicolin, we performed a 'staged depletion' experiment (*Figure 2B*). We synchronised cells first in G1 phase with Lovastatin (*Javanmoghadam-Kamrani and Keyomarsi, 2008*; *Rao et al., 1999*), and 8 hr after release from Lovastatin added Aphidicolin to induce replication stress (*Figure 2B*, upper timeline). The mitosis-specific CDK inhibitor RO-3306 (*Vassilev et al., 2006*) was simultaneously added, to induce a temporary G2 arrest and prevent cells from proceeding into mitosis. After 28 hr, Aphidicolin and RO-3306 were removed to allow release, and then 4 hr later cells were plated for CFA measurement of cell viability (*Vassilev et al., 2006*). Synchronisation timings were optimised using flow cytometry analysis of cell cycle progression (*Figure 2C*, *Figure 2—figure supplement 2B*).

Within the above synchronisation procedure, we either depleted RIF1 during the S phase Aphidicolin treatment period and re-expressed it for the recovery period (condition I, *Figure 2B*), or expressed RIF1 during the S phase treatment period and depleted it for the recovery period (condition II, *Figure 2B*). We included control samples where RIF1 was either expressed or depleted throughout the entire experiment (*Figure 2D*, RIF1+ and RIF1-). Cells depleted of RIF1 only during the replication stress treatment period (condition I) showed sensitivity similar to the RIF1-depleted (RIF1-) condition, displaying a mean surviving fraction of 33% (data from two biological replicates shown in *Figure 2—figure supplement 2C*). Cells depleted of RIF1 only during the recovery period (condition II) also showed high sensitivity, again similar to RIF1- with a mean surviving fraction of 26% (*Figure 2D*). These results indicate that molecular functions of RIF1 occurring both during and after a stressed S phase are important for survival. The cell death observed did not reflect an

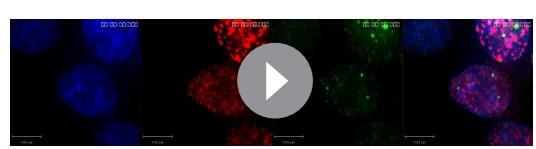

**Video 1.** mClover-tagged RIF1 from early to late S phase. Time-lapse imaging video of unsynchronised HCT116 mAC-RIF1 mCherry-PCNA cells transitioning from early to late S phase. Blue represents DNA (DAPI stain), red represents PCNA (mCherry-PCNA) and green represents mAC-RIF1 (mClover).

https://elifesciences.org/articles/58020#video1

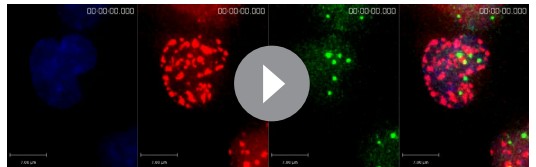

**Video 2.** mClover-tagged RIF1 from late S to the following G1 phase. Time-lapse imaging video of unsynchronised HCT116 mAC-RIF1 mCherry-PCNA cells transitioning from late S to the following G1 phase. Blue represents DNA (DAPI stain), red represents PCNA (mCherry-PCNA) and green represents mAC-RIF1 (mClover).

https://elifesciences.org/articles/58020#video2

effect of RO-3306, since cells expressing or lacking RIF1 showed no difference in RO-3306 sensitivity (data not shown).

We performed similar conditional depletion experiments in asynchronous cell populations (as outlined in *Figure 2—figure supplement 2D*), and observed comparable results (*Figure 2—figure supplement 2E*), in that depletion of RIF1 either during or after Aphidicolin treatment led to sensitivity.

To summarise, these results imply that RIF1 must be present during both treatment and recovery to protect cells from the effects of replication stress induced by Aphidicolin. The observation that RIF1 function remains important after a stressed replication period to promote cell survival is consistent with its highly dynamic pattern of localisation through late cell cycle stages (*Figure 1G,H*), suggesting that RIF1 operates in chromosome maintenance processes occurring outside of S phase.

## Only the RIF1-Long splice isoform protects cells from replication stress

The RIF1 messenger RNA undergoes alternative splicing resulting in expression of 'Long' and 'Short' protein isoforms that we name RIF1-L and RIF1-S. RIF1-S lacks 26 amino acids corresponding to exon 31 (*Figure 3A*, exon 31 shown in red; see Materials and methods for Exon designation). We confirmed that transcripts encoding both the isoforms are widely expressed in various tissues in vivo (*Figure 3—figure supplement 1*), and are present in U2OS and HEK293 cell lines (data not shown). Although they were mentioned as showing differential expression in cancer cells (*Xu and Blackburn, 2004*), distinct functions of the two isoforms have not previously been examined. We constructed cell lines expressing only mAC-RIF1-L or only mAC-RIF1-S, by inserting at the 3' end of Exon 29 a 'pre-spliced' cDNA construct consisting of Exons 30–35 including or excluding Exon 31 (*Figure 3A*). Clones were selected where both copies of the mAC-RIF1 gene contained the insertion. Western analysis confirmed that these mAC-RIF1-L and mAC-RIF1-S constructs encoded proteins expressed at levels similar to parental mAC-RIF1 (*Figure 3B*, lanes 2 and 3).

We found that whilst the mAC-RIF1-L cell line showed resistance to Aphidicolin very similar to that of the parent mAC-RIF1 cells (*Figure 3C*, *Figure 3—figure supplement 2*, black and red lines), the mAC-RIF1-S isoform in contrast conferred little protection against drug, producing sensitivity similar to that of cells lacking RIF1 altogether (*Figure 3C*, *Figure 3—figure supplement 2*, grey and dotted lines). This result suggests that only RIF1-L can protect cells from replication stress caused by Aphidicolin, and that RIF1-S is ineffective in this role.

To confirm this finding in a different cell line, we used HEK293-derived stable cell lines, containing siRIF1-resistant cDNA constructs encoding either GFP-RIF1-L or GFP-RIF1-S expressed under DOX control. Treatment of these cell lines with siRIF1, followed by DOX induction, allows replacement of endogenous RIF1 with either its Long or Short isoform (*Figure 3—figure supplement 3A,B and C*; *Hiraga et al., 2017*). We found that also in this cell line RIF1-L was able to protect against Aphidicolin treatment, whilst RIF1-S could not (*Figure 3D*, *Figure 3—figure supplement 3D and E*, red and grey lines). Similarly, in the same cell line RIF1-L was able to protect cells from Hydroxyurea treatment whilst RIF1-S could not (*Figure 3—figure supplement 3F*).

We considered molecular roles through which RIF1-L and RIF1-S might confer differing replication stress sensitivity. We found that both isoforms however appear equally functional in the previously described mechanisms through which RIF1 affects DNA replication. The isoforms are equally effective in preventing hyperphosphorylation of the MCM complex (*Figure 3E*, *Figure 3—figure supplement 4A*) and protecting nascent DNA at blocked replication forks (*Figure 3F*). Nor did we observe significant differences between cells expressing RIF1-S or RIF1-L in replication fork speed (as assessed by nascent DNA tract length; *Figure 3—figure supplement 4B*), bulk DNA synthesis levels (as measured by flow cytometry; *Figure 3—figure supplement 4C*), or replisome density (based on PCNA loading on chromatin; *Figure 3—figure supplement 4D*). RIF1-S therefore can apparently fulfil the known functions of RIF1 during DNA replication, and yet cannot safeguard cells from Aphidicolin treatment. These observations imply that responding to replication stress demands a further RIF1-mediated mechanism to promote cell survival, probably one that operates after the period of stress (*Figure 2*) and that specifically requires RIF1-L (*Figure 3C,D*).

In its functions in origin repression and nascent DNA protection during S phase, RIF1 acts as a PP1 substrate-targeting subunit (*Garzón et al., 2019*; *Hiraga et al., 2017*; *Hiraga et al., 2014*; *Mukherjee et al., 2019*). To further test for separability of the effect of RIF1 in replication stress survival from its known roles in replication control, we investigated whether PP1 interaction is essential for RIF1-L to protect against Aphidicolin treatment. We used the HEK293 cell line expressing a

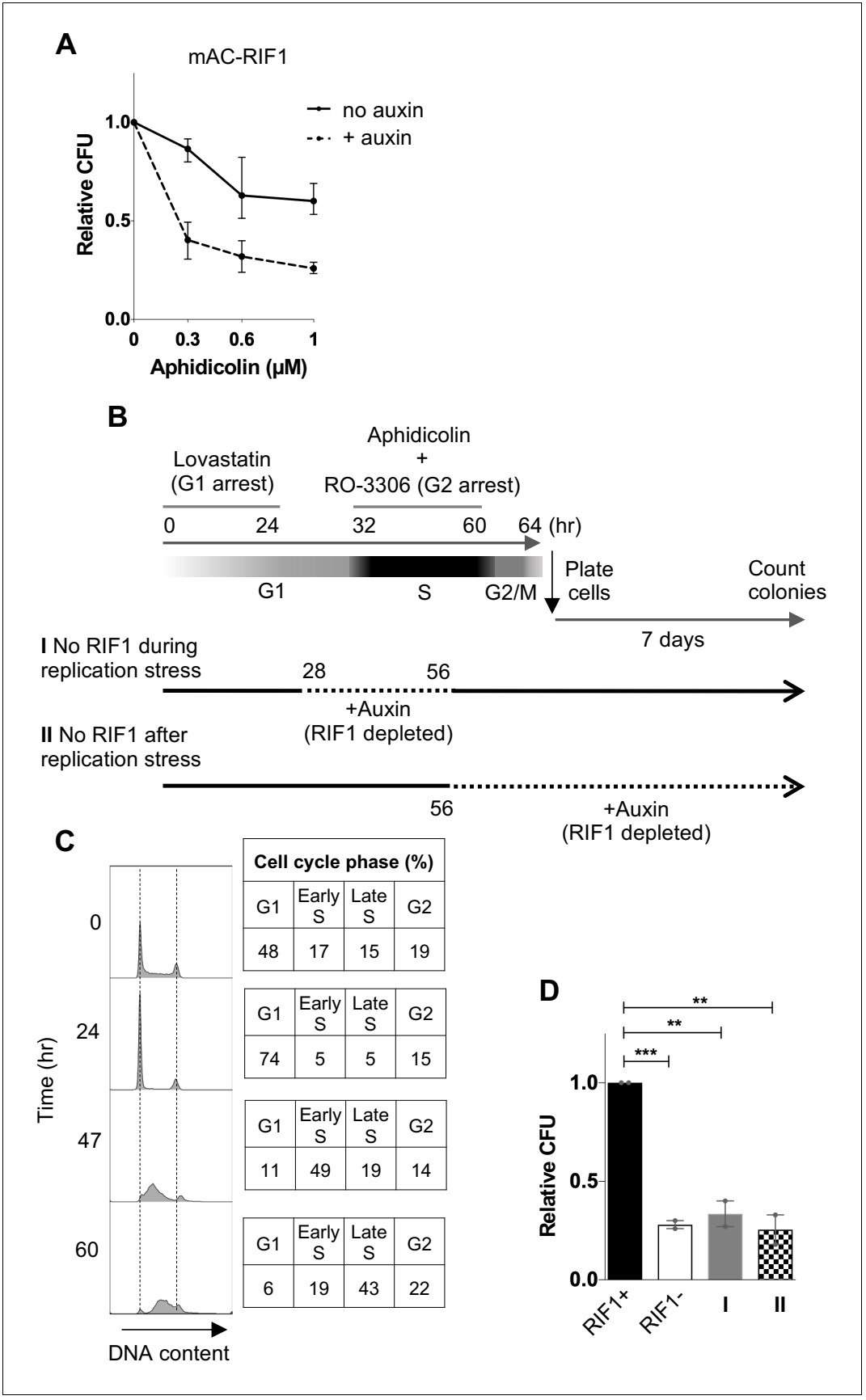

**Figure 2.** RIF1 is essential during and after drug treatment to protect against effects of replication stress. (**A**) CFA testing Aphidicolin sensitivity of HCT116 mAC-RIF1 cells depleted for RIF1. Cells were treated with DOX and Auxin for 48 hr before seeding at low density and treatment with Aphidicolin at the concentrations indicated. 'No Auxin' cells were treated with DOX and Auxinole. After Aphidicolin removal, medium was replaced with DOX + Auxin medium for '+ Auxin' cells or DOX + Auxinole medium for 'No Auxin' cells. Plot shows mean and range from three biological replicates, each plated in technical triplicate. (**B**) Procedure for testing effect of depleting RIF1 during or after Aphidicolin treatment in synchronised cultures. HCT116 mAC-RIF1 cells were incubated with Lovastatin to arrest cells in G1. After 24 hr, cells were released from G1 arrest with Mevalonic Acid. 4 hr into release (28 hr from the initial addition of Lovastatin), RIF1 depletion was induced in 'condition I' cells by addition of DOX and Auxin. 8 hr into release (32 hr), 1 μM Aphidicolin was added (to both cultures) to induce replication stress and simultaneously, RO-3306 was added to arrest cells in G2 phase. 24 hr later (at 56 hr), in 'condition I' cells RIF1 was re-expressed by removal of DOX and Auxin and addition of Auxinole. Also at 56 hr, in 'condition II' cells RIF1 depletion was induced by addition of DOX and Auxin. 4 hr later (60 hr), both cell cultures were released from RO-3306. After 4 hr (64 hr), cells were seeded in triplicate at 250/well in 6-well plates (condition I in medium containing DOX + Auxinole and condition II in medium containing DOX + Auxin) then incubated for 7 days after which colonies were counted. mAC-RIF1 depletion and re-expression data is shown in *Figure 2—figure supplement 1*. (**C**) Cell cycle progression in HCT116 mAC-RIF1 cells during the procedure shown in B. Synchronisation was performed as in B using 1 μM Aphidicolin. Histograms show distribution of cellular DNA content at the time indicated. Tables show % of cells in each cell cycle phase at the time indicated. (**D**) Effect on colony formation efficiency of mAC-RIF1 depletion procedures with 1 μM Aphidicolin treatment: 'condition I' and 'condition II' cells were treated according to the procedure in B. RIF1+ and RIF1- correspond to control mAC-RIF1 cells in which RIF1 was either expressed or depleted throughout the procedure. Values shown are normalised to the RIF1+ control. Plot shows the mean and range from two biological replicates, each plated in technical triplicate. **p<0.01; ***p<0.001. See *Figure 2—figure supplement 2C* for individual plots.

The online version of this article includes the following figure supplement(s) for figure 2:

**Figure supplement 1.** mAC-RIF1 depletion and re-expression.

**Figure supplement 2.** In asynchronous cycling cells, RIF1 is essential during and after drug treatment to protect against effects of replication stress.

---

version of RIF1-L mutated at the PP1 interaction motifs to prevent PP1 interaction (*Figure 3G,H*; *Hiraga et al., 2017*). This 'RIF1-L-pp1bs' protein was almost as effective as wild-type RIF1-L in conferring resistance to Aphidicolin (*Figure 3I*, *Figure 3—figure supplement 5A*), indicating that RIF1-L acts largely independent of PP1 function in protecting cells from the effects of Aphidicolin. This independence from PP1 reinforces the evidence that the function of RIF1 in protecting from replication stress is distinct from its previously known roles in replication control, which do require PP1.

## RIF1-Long and -Short isoforms do not appear to differ in mitotic functions

We considered other routes through which RIF1 might promote survival after Aphidicolin treatment, focusing on events occurring after the replication stress period itself. Chromosomal bridge structures have been reported to form after replication stress—in particular UFBs, believed to correspond to sites of under-replicated DNA that become single-stranded when chromosomes separate (*Chan et al., 2009*). UFBs do not stain with DAPI (which recognises double-stranded DNA). RIF1 acts at UFBs (*Hengeveld et al., 2015*), so we tested whether the RIF1-L and RIF1-S isoforms show differential recruitment to these sites or affect their rate of formation. UFB formation is induced by treatment with the topoisomerase II inhibitor ICRF-193 (*Figure 4Ai*,ii). We found no clear difference in rates of UFB formation in the HCT116 RIF1-L or RIF1-S cells after ICRF-193 treatment (*Figure 4Aii*, left), and both RIF1-L and RIF1-S were localised at the UFBs formed (*Figure 4Ai*, and *Figure 4Aii*, right). Treatment of these HCT116-based cell lines with Aphidicolin surprisingly did not induce UFBs, but rather increased the incidence of anaphase chromosome bridges (defined as bridge structures that stain with DAPI) (*Figure 4Ai*, Bi). These anaphase bridges were formed at similar rates in the RIF1-L and RIF1-S-expressing cells (*Figure 4Bii*, left) and both RIF1-L and RIF1-S were localised to these anaphase bridges (*Figure 4Bi*, Bii, right). Overall therefore, both RIF1 isoforms can localise effectively to both anaphase bridge and UFB structures; and the loss of RIF1 does not greatly affect the rate at which such bridge structures occur. The results of this analysis were consistent with observations made by Imagestream flow cytometry, where analysis of untreated cell lines revealed

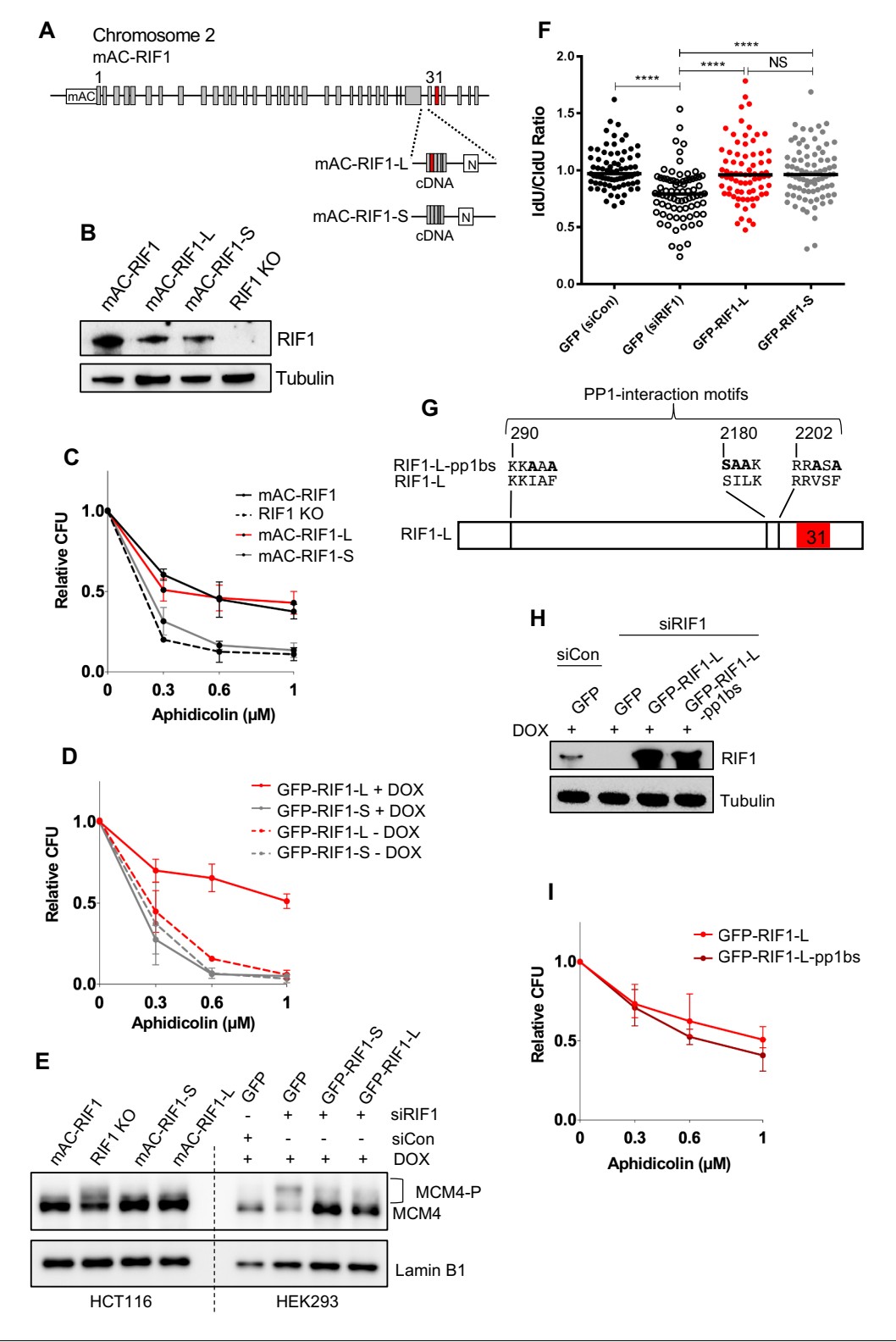

**Figure 3.** RIF1-L promotes resistance to Aphidicolin treatment but RIF1-S cannot. (**A**) RIF1 exon structure and construction of HCT116 mAC-RIF1-L and mAC-RIF1-S cell lines. Constructs containing cDNA encoding the C-terminal portion of either RIF1-L or RIF1-S were inserted at the end of exon 29 as illustrated, by CRISPR-mediated integration of donor plasmids into the HCT116 mAC-RIF1 cell line. 'N' is Neomycin resistance gene. (**B**)

*Figure 3 continued on next page*

*Figure 3 continued*

Expression of RIF1 in HCT116 mAC-RIF1, mAC-RIF1-L, mAC-RIF1-S and RIF1 KO cell lines. Whole cell extracts were harvested for western blotting with anti-RIF1 antibody. Tubulin is shown as a loading control. (C) CFA comparing resistance to Aphidicolin of mAC-RIF1, mAC-RIF1-L, mAC-RIF1-S, and RIF1-KO cell lines. Plot shows the mean and range from two biological replicates. See *Figure 3—figure supplement 2* for individual plots. (D) CFA comparing effects of HEK293 GFP-RIF1-L and GFP-RIF1-S expression on Aphidicolin resistance. Endogenous RIF1 was depleted in all samples by siRIF1 treatment prior to ectopic RIF1-L/S induction by DOX addition. Plot shows the mean and range from two biological replicates. See *Figure 3—figure supplement 3D* for individual experiments. (E) Both RIF1-L and RIF1-S can counteract MCM4 hyperphosphorylation caused by depletion of endogenous RIF1. For HEK293 cells, 24 hr after transfection with either siCon or siRIF1, DOX was added to the culture medium. 24 hr later, chromatin-enriched protein fractions were prepared and analysed by western blotting with anti-MCM4 antibody. Hyperphosphorylated MCM4 protein (MCM4-P) is detected based on its retarded mobility as indicated by bracket. (F) DNA fibre assay to assess nascent DNA degradation. HEK293-derived cell lines were transfected with siCon or siRIF1 to deplete endogenous RIF1. The following day, expression of the stably integrated GFP, GFP-RIF1-L and GFP-RIF1-S was induced by addition of DOX. 2 days later cells were labelled with CldU then IdU, followed by treatment with 2 mM hydroxyurea for 4 hr, after which the IdU:CldU ratio was measured in fibre analysis. Black bars represent median value. 75 forks were analysed per sample. NS: not significant; ****p≤0.0001. (G) Illustration of RIF1 PP1-interaction motifs. To prevent PP1 interaction, critical residues in all three potential PP1 interaction motifs were replaced with alanine, creating a RIF1-pp1bs allele as described (*Hiraga et al., 2017*). (H) Expression of GFP-RIF1-pp1bs in HEK293 Flp-In T-Rex cells. 48 hr after transfection with siRIF1, DOX was added to the culture medium. After 24 hr, expression was assessed by western blotting with anti-RIF1 antibody. Tubulin is shown as a loading control. (I) CFA comparing effect of GFP-RIF1-L and GFP-RIF1-L-pp1bs expression on Aphidicolin resistance. The CFA was carried out as in (D). Plot shows the mean and range from three biological replicates, each carried out in technical triplicate. See *Figure 3—figure supplement 5* for individual plots.

The online version of this article includes the following figure supplement(s) for figure 3:

**Figure supplement 1.** RIF1 splice variants are expressed in vivo.

**Figure supplement 2.** HCT116 mAC-RIF1-L promotes resistance to Aphidicolin treatment but mAC-RIF1-S cannot.

**Figure supplement 3.** In HEK293-derived cells RIF1-L promotes resistance to Aphidicolin treatment but RIF1-S cannot.

**Figure supplement 4.** Replication fork speed, bulk DNA synthesis levels and replisome density are similar in RIF1-L and RIF1-S.

**Figure supplement 5.** RIF1-L resistance to Aphidicolin treatment is largely independent of PP1 interaction.

chromosome bridge structures with localised RIF1 in 9 of 35 mitotic RIF1-L cells, and 8 of 38 mitotic RIF1-S cells.

We also tested for any effect of RIF1 in other phenotypes caused by replication stress, including chromatin breaks (monitored as breaks or gaps in specific loci of condensed mitotic chromosomes in metaphase spreads) or MiDAS (Mitotic DNA Synthesis). However, neither chromatin breaks nor MiDAS were significantly elevated in cells lacking RIF1 after Aphidicolin treatment (*Figure 4—figure supplement 1A,B*).

Overall, these investigations provided no evidence that the deficiency of the RIF1-S isoform in protecting cells against Aphidicolin is caused by its failure to function in mechanisms that manage under-replicated DNA during mitosis.

## RIF1-long promotes 53BP1 nuclear body formation in G1 phase

A further consequence of replication stress is the formation of large 53BP1 nuclear bodies (NBs) in the subsequent G1 phase, which protect unreplicated DNA damaged by chromosome breakage at mitosis (*Lukas et al., 2011*; *Moreno et al., 2016*). We examined the formation of 53BP1 NBs 12 hr after removal of Aphidicolin in our cells expressing only mAC-RIF1-L, only mAC-RIF1-S, or neither isoform. We limited our analysis to G1 phase cells by counting only those that were cyclin A2-negative (*Figure 5—figure supplement 1B*). The parental (mAC-RIF1) cell line showed an elevated fraction of cells with multiple large 53BP1 NBs (*Figure 5A* top row, 5B black bars and *Figure 5—figure supplement 1C* top left panel), as expected. RIF1 was often colocalised with these bodies (*Figure 5A* top row and 5C). In contrast, a reduced number of 53BP1 NBs were observed in RIF1-KO cells (*Figure 5A* bottom row, 5B open bars, *Figure 5—figure supplement 1C* bottom right

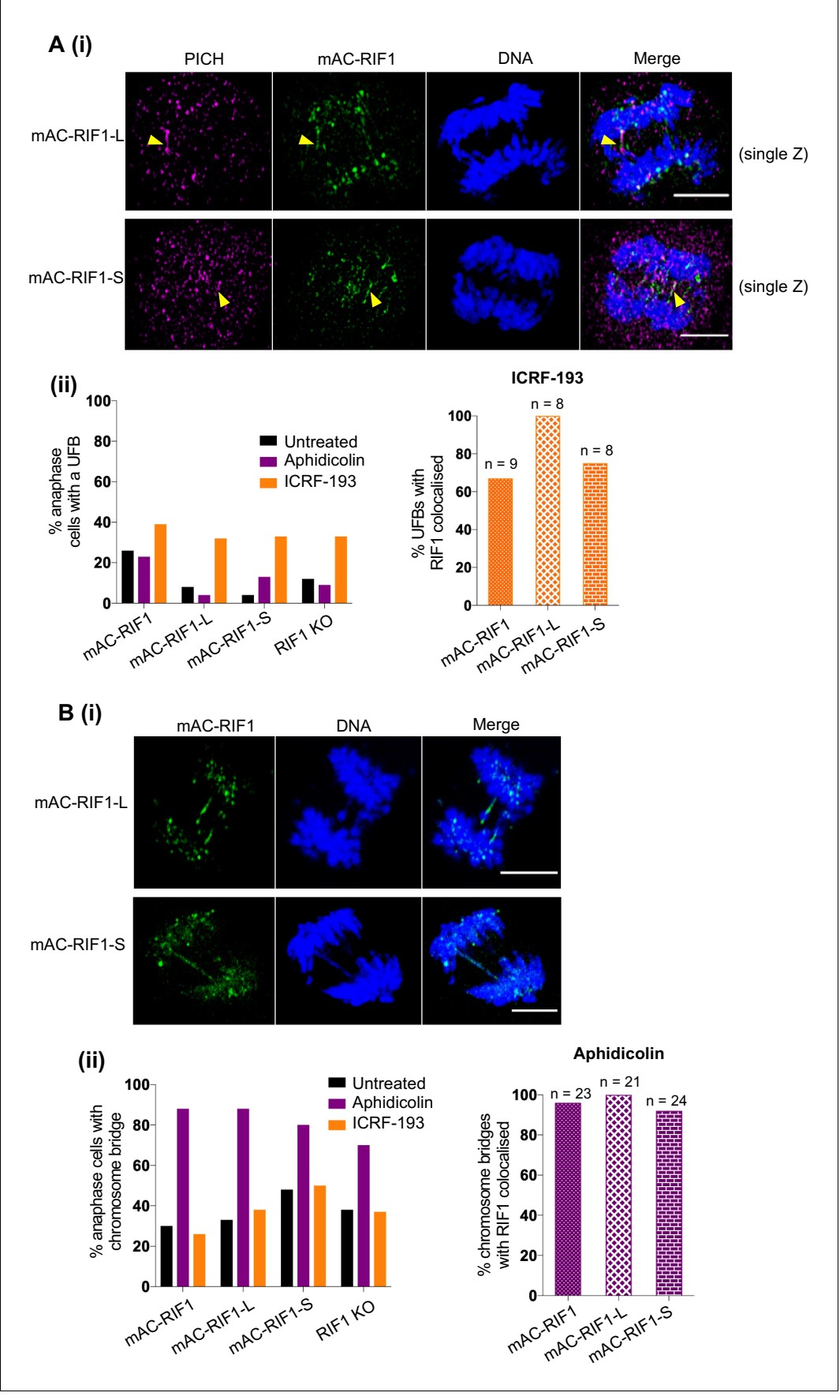

**Figure 4.** Both RIF1-L and RIF1-S localise to replication stress-induced chromosome bridges. (**A**) (i) Representative images showing PICH and mAC-RIF1 localised at UFBs induced by ICRF-193 treatment of HCT116 cells containing only RIF1-L (upper panels) or RIF1-S (lower panels) isoforms. Yellow arrowheads indicate UFBs. Scale bars = 5 μm. (ii) Left: Fraction of anaphase HCT116 cells with at least one UFB after treatment with Aphidicolin or ICRF-193. At least 23 anaphase cells were imaged for each sample. Right: % of ICRF-193-induced UFBs with colocalised RIF1. Cells were treated with 1 μM Aphidicolin for 24 hr then released for 12 hr before mitotic enrichment and fixation, or with 0.1 μM ICRF-193 for 30 min before mitotic enrichment and fixation. (**B**) (i) Representative images showing mAC-RIF1 at localised at chromosome bridges induced by Aphidicolin treatment of HCT116 cells containing only RIF1-L (upper panels) or RIF1-S (lower panels) isoforms. Scale bars = 5 μm. (ii) Left: Fraction of anaphase HCT116 cells with a chromosome bridge after treatment as described in A(ii). At least 23 anaphase cells were imaged for each sample. Right: % of Aphidicolin-induced chromosome bridges with colocalised RIF1.

The online version of this article includes the following figure supplement(s) for figure 4:

**Figure supplement 1.** Cells lacking RIF1 show neither elevated chromatin breaks nor altered MiDAS after Aphidicolin treatment.

---

panel), demonstrating that RIF1 contributes to the formation of 53BP1 bodies after replication stress. Examining the cell lines expressing only Long or Short RIF1 isoforms, we found that mAC-RIF1-L localised normally with 53BP1 bodies, and supported their formation at a near normal rate (*Figure 5A* second row, 5B,C red bars and *Figure 5—figure supplement 1C* top right panel). In contrast, the number of 53BP1 NBs formed in mAC-RIF1-S cells was similar to that in RIF1-KO cells, and mAC-RIF1-S showed reduced co-localisation with the 53BP1 bodies (*Figure 5A* third row, 5B,C grey bars and *Figure 5—figure supplement 1C* bottom left panel). The reduced rate of 53BP1 NB formation in mAC-RIF1-S cells was not due to reduced abundance of 53BP1 in mAC-RIF1-S when compared to mAC-RIF1-L cells (*Figure 5—figure supplement 1A*). The two RIF1 isoforms therefore differ in their effectiveness in promoting 53BP1 nuclear body formation following replication stress, with RIF1-L but not RIF1-S functional in this role. Since 53BP1 nuclear bodies are known to protect DNA damaged as a consequence of Aphidicolin treatment (*Lukas et al., 2011*), the defect in 53BP1 body formation in the absence of RIF1-L is likely to be an important factor in the replication stress sensitivity of RIF1-deficient cells, and may explain the isoform specificity of the replication stress protection function of RIF1.

To test in a different cell line whether RIF1 contributes to the number of 53BP1 nuclear bodies formed, we examined U2OS cells depleted of RIF1, 12 hr after release from a 24 hr Aphidicolin treatment (*Figure 5D*, *Figure 5—figure supplement 1D*). We found that RIF1 depletion caused a reproducible reduction in the number of NBs, similar to that observed in our HCT116-derived cell lines.

TopBP1 has been reported to be involved in the transition of under-replicated DNA sites damaged during mitosis into 53BP1 NBs in the following G1 phase (*Pedersen et al., 2015*). We therefore tested for involvement of TopBP1 in the stimulation of 53BP1 NB formation by RIF1-L. We found that depleting TopBP1 substantially reduced the rate of 53BP1 NB colocalisation with RIF1-L (*Figure 5—figure supplement 2A,B*). We conclude that TopBP1 is needed for effective localisation of RIF1-L with 53BP1 NBs, potentially enabling the specific function of RIF1-L in recovery from Aphidicolin treatment.

## Discussion

In investigating mechanisms through which RIF1 protects against interruption to replication, we tested RIF1 isoforms and found that RIF1-L is able to protect against replication stress whilst RIF1-S cannot. This deficiency of RIF1-S function was initially surprising, since RIF1-S appears competent to fulfil the known functions of RIF1 in DNA replication management—in particular RIF1-S is able to support replication licensing and control MCM phosphorylation (*Hiraga et al., 2017* and *Figure 3E*), and to protect against nascent DNA degradation (*Figure 3F*). Consistently however, all of these known functions of RIF1 in replication control (promotion of replication licensing, control of MCM phosphorylation, and nascent DNA protection) depend on PP1 recruitment by RIF1 (*Garzón et al., 2019*; *Hiraga et al., 2017*; *Hiraga et al., 2014*); whilst we find that protection against replication stress does not require PP1 interaction (*Figure 3I*, *Figure 3—figure supplement 5*), again suggesting that the role of RIF1 in protecting from replication stress might involve additional mechanisms.

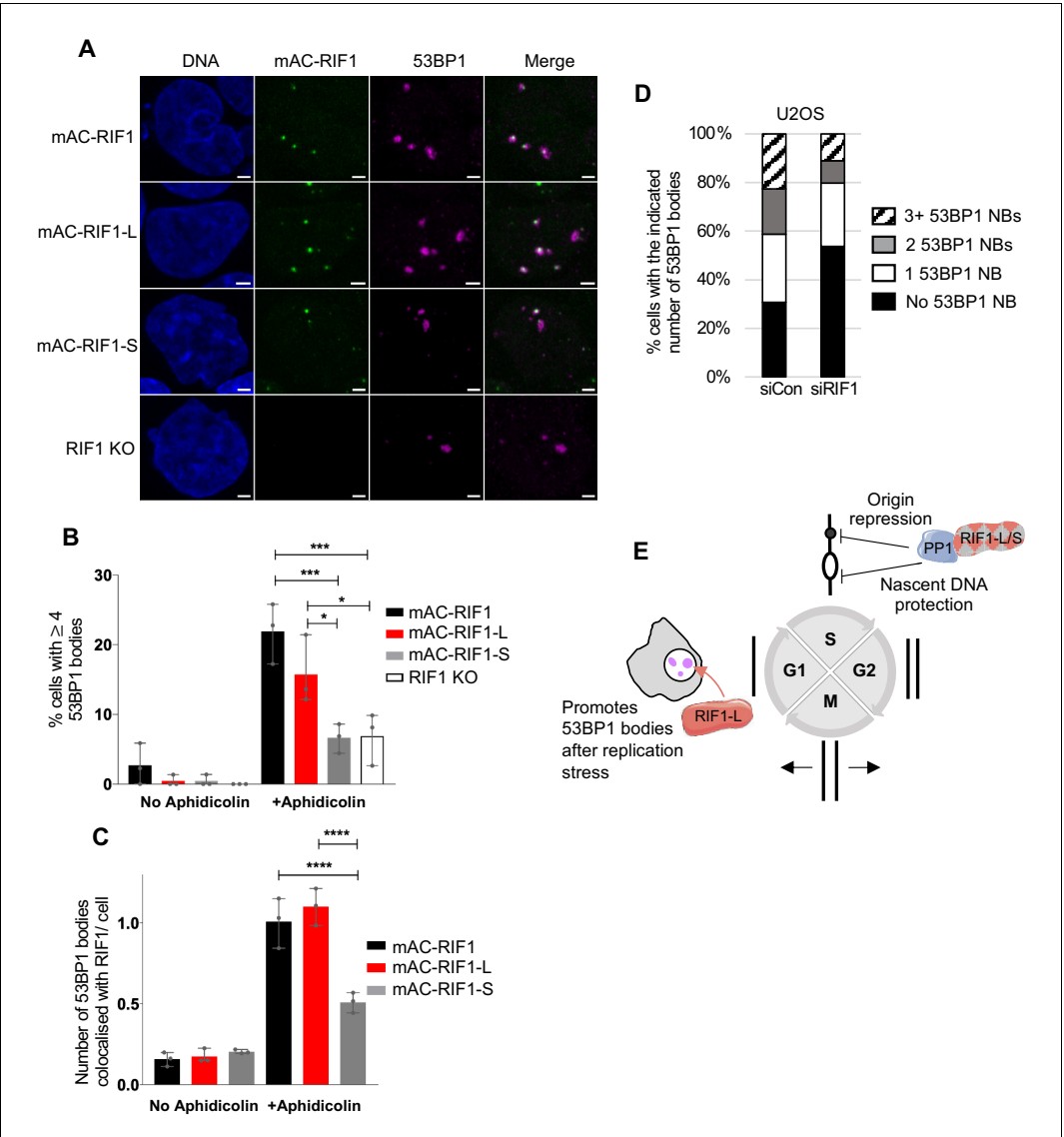

**Figure 5.** RIF1-L preferentially localises to 53BP1 protective nuclear bodies and promotes their formation. (A) Representative images showing 53BP1 nuclear bodies and RIF1 foci in HCT116 cell lines. Cells were treated with 1 μM Aphidicolin for 24 hr then released for 12 hr before fixation. In total, a minimum of 132 cells were analysed per sample. Scale bar = 2 μm. (B) Number of cells with 4 or more 53BP1 nuclear bodies in cells either untreated or treated with Aphidicolin as in A. Plot shows the mean and range from three independent experiments. *p<0.05; ***p<0.001. See *Figure 5—figure supplement 1C* for plots showing distribution of 53BP1 foci per cell. (C) Number of 53BP1 nuclear bodies colocalised with RIF1 foci per cells in untreated and Aphidicolin-treated HCT116 cells. Plot shows the mean and range from three biological replicates. ****p<0.0001. (D) Mean distribution of 53BP1 NBs in U2OS cells treated as in A from three biological replicates. In total, a minimum of 140 cells were analysed per sample. See *Figure 5—figure supplement 1D* for individual plots. (E) Illustration of RIF1-L and -S isoform functions.

The online version of this article includes the following figure supplement(s) for figure 5:

**Figure supplement 1.** Distribution of 53BP1 nuclear bodies after Aphidicolin treatment.

**Figure supplement 2.** Colocalisation of 53BP1 with RIF1-L requires TopBP1.

Testing the effects of conditional depletion in synchronised cultures revealed that RIF1 is still needed after a period of replication stress to guard against toxicity, implying that protection from Aphidicolin involves a further function of RIF1. We therefore investigated whether RIF1 operates in post-S phase replication stress response pathways. RIF1 was reported to act at chromosome bridge

structures formed during mitosis after replication stress, so we tested whether the RIF1-L and RIF1-S isoforms function differently in this role. However, our investigations provided no evidence for differential roles of RIF1-L and RIF1-S at UFBs or anaphase bridges (*Figure 4*). Nor did we find any role for RIF1 in determining rates of MiDAS or chromatin breaks (*Figure 4—figure supplement 1A,B*). Our finding that RIF1 does not affect chromatin breaks contrasts with the results of *Debatisse and Rosselli, 2019*, who observed that RIF1 depletion suppresses fragile site breakage in human lymphoblastoid cells. Debatisse and Rosselli interpreted this suppression as due to earlier replication of fragile site DNA in lymphoblastoid cells in the absence of RIF1. The lack of any effect in our HCT116-based line may reflect cell line-specificity of RIF1 effects on replication timing (*Gnan et al., 2020*; *Klein et al., 2019*).

Rather than a differential role of the RIF1 isoforms during mitosis, our investigation revealed a role for RIF1-L in promoting the assembly of G1 phase 53BP1 nuclear bodies (*Figure 5E*). Although in DNA repair RIF1 is generally thought to act downstream of 53BP1, RIF1 was recently shown to contribute to the correct substructure of 53BP1 ionizing radiation-induced foci (IRIF) that form at radiation-induced double-strand breaks (*Ochs et al., 2019*). Ochs et al did not report the number of 53BP1 IRIF foci formed in the absence of RIF1, or test effects of the RIF1 isoforms. The factors important for the larger, replication stress-induced G1 phase 53BP1 nuclear body (NB) structures are less well understood, and there has been no investigation of how RIF1 affects NB formation. The role we have discovered for RIF1-L could be related to the effect described by Ochs et al for RIF1 in 53BP1 IRIF organisation, but in RIF1-S cells we found no evident reduction in IRIF formation after DSB-inducing treatment, in contrast to the clear defect in replication stress-induced G1 phase 53BP1 NB formation (*Figure 5*). This difference suggests that the pathway through which RIF1 organises IRIF substructure (Ochs et al) may differ from the pathway through which RIF1-L promotes 53BP1 NB formation (this study). Factors that have been implicated upstream of 53BP1 NB formation include phospho-ATM and RNF168 (*Lukas et al., 2011*), but an initial investigation of these factors in RIF1-L and RIF1-S cells revealed no striking differences.

Our finding that RIF1-L but not RIF1-S can function in promoting 53BP1 body formation potentially explains the specific requirement for RIF1-L in protecting against replication stress, since 53BP1 NB assembly represents an important step in correct handling of stress-associated damage to enable ongoing proliferation. RIF1-L may act to directly promote 53BP1 NB assembly, or possibly assist with the transit of damaged sites through to G1 phase to allow such protective bodies to form. Presently it is unclear how the apparently small difference of RIF1-L from the RIF1-S isoform (the inclusion of just 26 amino acids) specifically supports 53BP1 nuclear body formation. Interestingly however, this region contains the motif 'SPKF' at amino acids 2265–2268, with the S-2265 residue identified as phosphorylated in high-throughput proteomic studies (*Schweppe et al., 2014*; *Sharma et al., 2014*) and in our own proteomics data. This sequence conforms to a 'phosphoSPxF' BRCT interaction motif, raising the possibility that RIF1-L interacts with one of the several BRCT domain-containing proteins involved in chromosome maintenance (which include BRCA1, TopBP1 and 53BP1 itself) to effect its role in promoting 53BP1 NB formation and protecting from Aphidicolin. However, the physical interaction of RIF1 with 53BP1 does not appear to be mediated through a BRCT domain-mediated interaction (*Callen et al., 2013*). We found that TopBP1 is important to enable the observed association of RIF1-L with 53BP1 NBs. However, co-immunoprecipitation analysis of RIF1-L following Aphidicolin treatment did not identify any interaction with TopBP1 (data not shown), so the molecular mechanism through which RIF1-L promotes 53BP1 NB formation remains an open question.

The findings described here suggest that RIF1 contributes to recognising under-replicated and unrepaired sites for special protection and handling later in the chromosome cycle—in particular for delayed replication that guards against unscheduled recombinational repair to prevent the formation of pathological intermediates, as recently described (*Spies et al., 2019*). Overall, this study highlights the multifunctional role of RIF1 in ensuring chromosome maintenance to promote the survival and proliferation of cells after replication stress, emphasising the importance of RIF1 for determining response to replication-inhibiting chemotherapeutic drugs.

# Materials and methods

## Key resources table

| Reagent type (species) or resource | Designation | Source or reference | Identifiers | Additional information |
|---|---|---|---|---|
| Cell line (*Homo sapiens*) | HEK293 Flp-In T-REx 293 derived cell lines: GFP, GFP-RIF1-L, GFP-RIF1-L-pp1bs, GFP-RIF1-S | This study and *Hiraga et al., 2017* | | |
| Cell line (*Homo sapiens*) | HCT116 OsTIR1 derived cell lines: mAC-RIF1, mAC-RIF1 mCherry-PCNA, mAC-RIF1-L, mAC-RIF1-S | This study | | |
| Cell line (*Homo sapiens*) | HCT116 RIF1 KO | This study | | |
| Cell line (*Homo sapiens*) | U2OS | ATCC | RRID:CVCL_0042 | |
| Transfected construct (human) | pcDNA5/FRT/TO-GFP-RIF1 derived plasmids: L, L-pp1bs | This study | | |
| Transfected construct (human) | pOG44 | *O'Gorman et al., 1991* | | Flp recombinase expression vector |
| tTransfected construct (human) | pX330-RIF1-N | This study | | CRISPR/Cas9 plasmid targeting exon one for construction of HCT116 mAC-RIF1 cell line |
| Transfected construct (human) | pmAC-RIF1 | This study | | Donor plasmid for construction of HCT116 mAC-RIF1 cell line |
| Transfected construct (human) | pX330-RIF1-29 | This study | | CRISPR/Cas9 plasmid targeting exon 29 for construction of HCT116 mAC-RIF1-L/S cell lines |
| Transfected construct (human) | pmAC-RIF1-L | This study | | Donor plasmid for construction of HCT116 mAC-RIF1-L cell line |
| Transfected construct (human) | pmAC-RIF1-S | This study | | Donor plasmid for construction of HCT116 mAC-RIF1-S cell line |
| Transfected construct (human) | pRIF1-KO | This study | | Donor plasmid for construction of HCT116 RIF KO cell line |
| Antibody | Rabbit polyclonal anti-RIF1 | Bethyl Laboratories | Cat#A300-568A RRID:AB_669806 | WB (1:5000) |
| Antibody | Rabbit polyclonal anti-GFP | Abcam | Cat#ab290 RRID:AB_303395 | WB (1:100,000) |
| Antibody | Rat monoclonal anti-Tubulin | Santa Cruz Biotechnology | Cat#sc-53030 RRID:AB_2272440 | WB (1:2000) |
| Antibody | Rabbit polyclonal anti-MCM4 | Abcam | Cat#ab4459 RRID:AB_304468 | WB (1:2000) |
| Antibody | Rabbit polyclonal anti-MCM2 phospho-S53 | Bethyl Laboratories | Cat#A300-765A | WB (1:2500) |
| Antibody | Mouse monoclonal anti-PCNA | Santa Cruz Biotechnology | Cat#sc-56 RRID:AB_628110 | WB (1:1000) |
| Antibody | Rabbit monoclonal anti-Lamin B1-HRP | Abcam | Cat#ab194109 | WB (1:5000) |
| Antibody | Rabbit polyclonal anti-TopBP1 | Bethyl Laboratories | Cat#A300-111A RRID:AB_2272050 | WB (1:5000) |
| Antibody | Rat monoclonal anti-CldU | Abcam | Cat#ab6326 | IF (1:100) |
| Antibody | Mouse monoclonal anti-IdU | Beckton Dickinson | Cat#347580 | IF (1:100) |
| Antibody | Mouse monoclonal anti-ssDNA | Merck Millipore | Cat#MAB3034 RRID:AB_94645 | IF (1:00) |

*Continued on next page*

Continued

| Reagent type (species) or resource | Designation | Source or reference | Identifiers | Additional information |
|---|---|---|---|---|
| Antibody | Goat polyclonal anti-Rat IgG Alexa Fluor 594 | Invitrogen | Cat#A11007 RRID:AB_141374 | IF (1:300) |
| Antibody | Goat polyclonal anti-Mouse IgG1 Alexa Fluor 488 | Invitrogen | Cat#A21121 RRID:AB_2535764 | IF (1:300) |
| Antibody | Goat polyclonal anti-Mouse IgG2a Alexa Fluor 350 | Invitrogen | Cat#A21130 RRID:AB_1500822 | IF (1:300) |
| Antibody | Rabbit polyclonal anti-53BP1 | Santa Cruz Biotechnology | Cat#sc22760 RRID:AB_2256326 | IF (1/500) |
| Antibody | Rabbit polyclonal anti-FANCD2 | Novus Biologics | Cat#NB100-182 RRID:AB_10002867 | IF (1/500) |
| Antibody | Goat polyclonal anti-BLM | Santa Cruz Biotechnology | Cat#sc7790 RRID:AB_2243489 | IF (1/500) |
| Antibody | Goat polyclonal anti-Rabbit IgG Alexa Fluor 594 | Invitrogen | Cat#A-11037 RRID:AB_2534095 | IF (1/1000) |
| Antibody | Donkey polyclonal anti-Goat Alexa 594 | Invitrogen | Cat#A-11058 RRID:AB_142540 | IF (1/1000) |
| Antibody | Alpaca/recombinant $V_H$H domain monoclonal GFP-Booster ATTO 488 | Chromotek | Cat#gba488 RRID:AB_2631386 | IF (1:000) |
| Antibody | Rabbit monoclonal anti-CyclinA2 Alexa Fluor 555 | Abcam | Cat#ab217731 | IF (1:000) |
| Antibody | Rabbit polyclonal anti-53BP1 | Novus Biologics | Cat#NB100-94 | IF and WB (1:1000) |
| Antibody | Mouse monoclonal anti-PICH | Sigma | Cat#04–1540 RRID:AB_10616795 | IF (1:00) |
| Antibody | Donkey polyclonal IgG anti-Mouse Alexa Fluor 647 | Abcam | Cat#ab150111 | IF (1:1000) |
| Recombinant DNA reagent | pcDNA5/FRT/TO-GFP-RIF1 derived plasmids:L, L-pp1bs | This study | | |
| Recombinant DNA reagent | pOG44 | *O'Gorman et al., 1991* | | Flp recombinase expression vector |
| Recombinant DNA reagent | pX330-RIF1-N | This study | | CRISPR/Cas9 plasmid targeting exon one for construction of HCT116 mAC-RIF1 cell line |
| Recombinant DNA reagent | pmAC-RIF1 | This study | | Donor plasmid for construction of HCT116 mAC-RIF1 cell line |
| Recombinant DNA reagent | pX330-RIF1-29 | This study | | CRISPR/Cas9 plasmid targeting exon 29 for construction of HCT116 mAC-RIF1-L/S cell lines |
| Recombinant DNA reagent | pmAC-RIF1-L | This study | | Donor plasmid for construction of HCT116 mAC-RIF1-L cell line |
| Recombinant DNA reagent | pmAC-RIF1-S | This study | | Donor plasmid for construction of HCT116 mAC-RIF1-S cell line |
| Recombinant DNA reagent | pRIF1-KO | This study | | Donor plasmid for construction of HCT116 RIF KO cell line |
| Sequence-based reagent | siRNA against RIF1 5' AGACGGTGCTCTATTGTTA 3' | Horizon Discovery | Cat#D-027983-02-0050 | |
| Sequence-based reagent | siRNA against Luciferase GL2 Duplex 5' CGTACGCGGAATACTTCGA 3' | Horizon Discovery | Cat#D-001100–01 | |
| Sequence-based reagent | siRNA against TopBP1 5' GGATATATCTT TGCGGTTTT 3' | Life Technologies | Cat#s21823 | |

*Continued*

| Reagent type (species) or resource | Designation | Source or reference | Identifiers | Additional information |
|---|---|---|---|---|
| Sequence-based reagent | Primers:<br>5′ – CTATGGAATTGAATGTAG GAAATGAAGCTAGC – 3′<br>5′ – ACCGAGCTCGGATCGAT CACCATGACGGCCAGGG – 3′<br>5′ – GCCGCGGATCCGAATTCTAA ATAGAATTTTCATGGGATGG – 3′<br>5′ –GCTACGTGATCCTGGGGACAG AAATCCTTTGGCTGAAGTGGTATTA TGCTTAGATTGTGTAGTAGGAGAAG – 3′<br>5′ –TCCCCAGGATCACGTAGCCCT AAATTTAAGAGCTCAAAGAAGT GTTTAATTTCAGAAATGGCCAAAG– 3′<br>5′ – GATCAGTTATCTATGCGGCCG – 3′ | Sigma | | Primers used to construct pcDNA5/ FRT/TO-GFP-RIF1-L |
| Sequence-based reagent | Primers:<br>5′- ccgggctgcaggaattcgatTAGGAG GGAGCGCGCCGCACGCGTG – 3′<br>5′ – ggcttttttcatggtggcgatCACCCT GAGGCCCGAACCGGAAGAG – 3′<br>5′-gctggtgcaggcgccggatccATGAC GGCCAGGGGTCAG AGtCCCCTCGCGCC – 3′<br>5′ – acggtatcgataagcttgatCTCT GGGTAGCCACATTTTCCCAAC – 3′ | Eurofins Genomics | | Primers used to amplify genomic DNA for the homology arms for the mAC-RIF1 donor plasmid pmAC-RIF1 |
| Sequence-based reagent | Primers:<br>5′- ccgggctgcaggaattcgatTA GGAGGGAGCGCGCCGCACGCGTG– 3′<br>5′ - tcgctgcagcccggggggatcGGGGGCT CTGACCCCTGGCCGTCATGTCGC– 3′<br>5′ – aagcttatcgataccgtcgaCTTTGGA AGACCCTTCTGCCTCCCATGGAG – 3′<br>5′ – acggtatcgataagcttgatCTCT GGGTAGCCACATTTTCCCAAC – 3′ | Eurofins Genomics | | PCR primers used to amplify genomic DNA for the homology arms for the RIF1 KO donor plasmid pRIF1-KO |
| Sequence-based reagent | Primers:<br>5′ – AAATCTCATCACCTGTTAATAAG – 3′ 5′ – acaagttaacaacaacaattCTAAA TAGAATTTTCATGGGATGGT– 3′ | Eurofins Genomics | | PCR primers used to amplify the C-terminal portion of either pcDNA5/FRT/TO-GFP-RIF1-L or pcDNA5/FRT/TO-GFP-RIF1 for donor plasmids pmAC-RIF1-L and pmAC-RIF1-S |
| Sequence-based reagent | Primers:<br>5′ – ATGCAgagctcGAAACAGA GAATGAGGGCATAACTA – 3′<br>5′ – ATGCAggtaccATTCATTC AACAAACTATGTGCAAG – 3′ | Eurofins Genomics | | PCR primers used to amplify the homology arms for donor plasmids pmAC-RIF1-L and pmAC-RIF1-S |
| Sequence-based reagent | Primers:<br>5′– GTCTCCTTTGGCTTCTCCGT–3′<br>5′–GATGTCAACTGGTGCCACAC–3′ | Sigma | | PCR primers used for cDNA analysis of RIF1 splice variants |
| Commercial assay or kit | Clarity ECL Blotting Substrate | Bio-Rad | Cat#1705061 | |
| Commercial assay or kit | RC-DC protein assay kit | Bio-Rad | Cat#5000121 | |
| Commercial assay or kit | In-Fusion HD Cloning kit | Clonetech | Cat#639646 | |
| Commercial assay or kit | Fugene HD | Promega | Cat#E2311 | |
| Commercial assay or kit | Click-iT EdU Imaging Kit | Invitrogen | Cat#C10340 (AF647) Cat#C10337 (AF488) | |

*Continued on next page*

*Continued*

| Reagent type (species) or resource | Designation | Source or reference | Identifiers | Additional information |
|---|---|---|---|---|
| Commercial assay or kit | Lightning-Link Rapid Alexa Fluor 647 Antibody Labeling Kit | Expedeon | Cat#336–0030 | |
| Commercial assay or kit | TissueScan, Human Normal cDNA Array | Insight Biotechnology | Cat#HMRT104 | |
| Chemical compound, drug | Lovastatin | Sigma | Cat#M2147 | |
| Chemical compound, drug | Auxinole | Bioacademia | Cat#30–001 | |
| Chemical compound, drug | Auxin | Sigma | Cat#I3750 | |
| Chemical compound, drug | Hydroxyurea | Sigma | Cat#H8627 | |
| Chemical compound, drug | Aphidicolin | Abcam | Cat#ab142400 | |
| Chemical compound, drug | Doxycycline | Sigma | Cat#D9891 | |
| Chemical compound, drug | Mevalonic Acid | Sigma | Cat#4667 | |
| Chemical compound, drug | RO-3306 | Sigma | Cat#SML0569 | |
| Chemical compound, drug | ICRF-193 | Sigma | Cat#I4659 | |
| Chemical compound, drug | Colcemid | Gibco | Cat#15212012 | |
| Chemical compound, drug | FcCyclePI/RNase | Molecular Probes | Cat# F10797 | |
| Chemical compound, drug | SiR-DNA | Spirochrome | Cat#SC007 | |
| Chemical compound, drug | Ibidi Mounting Medium | Ibidi | Cat#50001 | |
| Chemical compound, drug | Prolong Glass Mounting Medium | Invitrogen | Cat#15808401 | |
| Chemical compound, drug | 10% Neutral Buffered Formalin | Sigma | Cat#HT5012 | |
| Software, algorithm | Cell Profiler (3.1.8) | Broad Institute | https://cellprofiler.org RRID:SCR_007358 | |
| Software, algorithm | Zen Black | Zeiss | https://www.zeiss.com RRID:SCR_018163 | |
| Software, algorithm | Prism 7 | Graphpad Software | https://www.graphpad.com RRID:SCR_002798 | |
| Software, algorithm | Volocity | PerkinElmer | https://www.perkinelmer.com RRID:SCR_002668 | |
| Software, algorithm | SoftWoRx | Cytiva Life Sciences | https://www.cytiva lifesciences.com | |
| Software, algorithm | FlowJo | Beckton, Dickinson and Company | https://www.flowjo.com RRID:SCR_008520 | |
| Software, algorithm | FIJI | ImageJ Software | https://imagej.net/Fiji RRID:SCR_002285 | |
| Software, algorithm | ImageLab | Bio-Rad | https://www.bio-rad.com | |
| Other | Microscope: LSM880 + Airyscan | Zeiss | | |

*Continued on next page*

Continued

| Reagent type (species) or resource | Designation | Source or reference | Identifiers | Additional information |
|---|---|---|---|---|
| Other | Microscope: Axioplan 2 Epifluorescence | Zeiss | | |

## Cell lines

Stable HEK293 Flp-In T-REx GFP and GFP-RIF1-S cell lines were as described (*Escribano-Díaz et al., 2013*; *Hiraga et al., 2017*). Constructed using the same procedure, described briefly under 'HEK293 cell lines and culture conditions', were cell lines: HEK293 Flp-In T-REx GFP-RIF1-L and HEK293 Flp-In T-REx GFP-RIF1-L-pp1bs. Construction of the necessary plasmids is described in the next section.

The HCT116 mAC-RIF1 cell line was constructed as outlined below in 'HCT116 cell lines and culture conditions', using the system described (*Natsume et al., 2016*; *Yesbolatova et al., 2019*). HCT116 mAC-RIF1-L, HCT116 mAC-RIF1-S and HCT116 RIF1 KO cells were constructed as described below. HCT116 mAC-RIF1 mCherry-PCNA was constructed by introducing the mCherry-PCNA construct under control of the EF1 alpha promoter using the piggyBac system (*Yusa et al., 2011*).

For the HEK293-derived cell lines, presence of the Tet-repressor modification provides confirmation of cell line identity. For HCT116 OsTIR1-derived cells and HCT116 RIF1 KO cell lines, presence of the OsTIR1 modification confirms cell line identity. U2OS cells are from ATCC. Cell lines tested negative for mycoplasma.

## Plasmids used for cell line constructions

RIF1 is encoded on chromosome 2. The RIF1-Long isoform cDNA (RIF1-L; NCBI RefSeq NM_018151.4) encodes a 2,472-amino acid protein, whilst the short isoform (RIF1-S; RefSeq NM_001177663.1) lacks the 78-nucleotide stretch corresponding to exon 31 (*Xu and Blackburn, 2004*). In this study we designate the exon containing the RIF1 ATG start codon as 'exon 1', so that our 'exon 31' corresponds to 'exon 32' of RefSeq NM_018151.4.

The GFP-RIF1 constructs used in this study are based on pcDNA5/FRT/TO-GFP-RIF1 (*Escribano-Díaz et al., 2013*), which carries human RIF1-S cDNA with GFP fused at its N-terminus. To construct pcDNA5/FRT/TO-GFP-RIF1-L, a PCR fragment containing RIF1-S cDNA was amplified from pcDNA5/FRT/TO-GFP-RIF1 using primers SH593 and SH594, and cloned into pIRESpuro3 vector (linearised by EcoRV and EcoRI) using the In-Fusion HD cloning system, to create plasmid pSH1009. The NheI-NotI fragment of the plasmid pSH1009 was replaced using the In-Fusion HD system by two PCR fragments (amplified by SH572 and SH595 and SH596 and SH597 respectively), to construct pSH1011 which has RIF1-L cDNA. The NheI-PspOMI fragment of pcDNA5/FRT/TO-GFP-RIF1 was replaced by NheI-NotI fragment of the pSH1011 plasmid to construct pcDNA5/FRT/TO-GFP-RIF1-L. Construction of a GFP-RIF1-S-pp1bs plasmid was previously described (*Hiraga et al., 2017*). The GFP-RIF1-L-pp1bs construct was made following a similar strategy.

The plasmid pX330-U6-chimeric_BB-CBh-hSPCas9 from Feng Zhang (Addgene, 42230) (*Cong et al., 2013*) was used to construct the CRISPR/Cas9 vector for the guide RNA according to the protocol described (*Ran et al., 2013*). Donor plasmids were based on pBluescript and constructed as described (*Natsume et al., 2016*; *Yesbolatova et al., 2019*). Primers for amplification of the homology arms and cDNA from RIF1-L and RIF1-S are listed in the key resource table.

## HEK293 cell lines and culture conditions

HEK293-derived cell lines were cultivated in Dulbecco's Modified Eagle's Minimal medium supplemented with 10% foetal bovine serum (tetracycline-free), 100 U/ml penicillin, and 100 µg/ml streptomycin at 37°C with 5% $CO_2$.

To construct cell lines, pOG44 (*O'Gorman et al., 1991*) and pcDNA5/FRT/TO-based plasmids carrying the GFP-RIF1-L or GFP-RIF1-L-pp1bs gene were mixed in 9:1 molar ratio and used for transfection of Flp-In T-REx 293 cells (Invitrogen) with Lipofectamine 3000 reagents (Invitrogen). Transfections and hygromycin B selection of stably transfected cells were performed as described by the

manufacturer. Clones were tested for doxycycline-dependent induction of GFP fusion proteins by western blot and microscopy.

To assess the effect of ectopically expressing RIF1, cells were transfected with either Control siRNA or siRNA against human RIF1 (see key resource table for sequence). 2 days later, cells were split with addition of 1 µg/ml DOX then incubated for 24 hr to induce expression of GFP-RIF1 variant proteins. siRNA transfection was carried out using Lipofectamine RNAiMAX (Invitrogen). Synonymous base mutations in the ectopically expressed GFP-RIF1 constructs make them resistant to siRNA targeted against endogenous RIF1 (*Escribano-Díaz et al., 2013*). RIF1 expression was assessed by western blot using RIF1 and GFP antibodies.

## HCT116 cell lines and culture conditions
HCT116-derived cells were cultivated in McCoys 5A medium supplemented with 2 mM L-glutamine, 10% foetal bovine serum (tetracycline-free), 100 U/ml penicillin, and 100 µg/ml streptomycin at 37°C with 5% $CO_2$.

### HCT116 mAC-RIF1
To construct miniAID-mClover-fused RIF1 stable cell lines, HCT116 cells expressing the auxin-responsive F-box protein *Oryza sativa* TIR1 (OsTIR1) under the control of a Tet promoter were transfected using FuGENE HD (Promega) with a CRISPR/Cas9 plasmid (pX330-RIF1-N) targeting nearby the 1 st ATG codon of the *RIF1* gene (5'-TCTCCAACAGCGGCGCGAGGggg-3') together with a donor plasmid (pmAC-RIF1) based on pMK345 (*Yesbolatova et al., 2019*), that contains a cassette (hygromycin resistance marker, self-cleaving peptide P2A, and mAID–mClover; *Yesbolatova et al., 2019*) flanked by 500 bp of homology arms on each end. Two days after transfection cells were diluted in 10 cm dishes, to which 100 µg/mL of Hygromycin B Gold (Invivogen) was added for selection. After 10–12 days, colonies were picked for further selection in a 96-well plate. Bi-allelic insertion of the donor sequence was checked by genomic PCR. Clones taken forward for analysis of RIF1 function were selected based on near endogenous RIF1 levels of RIF1 and efficient AID-mediated degradation of RIF1.

To induce degradation of miniAID-mClover-fused RIF1, OsTIR1 expression was first induced by 0.2 µg/ml DOX added to the culture medium, to produce a functional SCF (Skp1–Cullin–F-box) ubiquitin ligase that directs degradation of an AID-tagged protein (*Natsume et al., 2016*; *Nishimura et al., 2009*). After 24 hr, 10 µM Auxin (indole-3-acetic acid) was added to the culture medium to promote the interaction of mAC-RIF1 with SCF-OsTIR1, driving ubiquitination and subsequent degradation of mAC-RIF1. To suppress premature degradation of RIF1 in the presence of DOX, 100 µM of the auxin antagonist Auxinole, which interferes with TIR1 activity, was added (*Hayashi et al., 2012*; *Yesbolatova et al., 2019*). In subsequent depletion experiments DOX was first added in the presence of Auxinole, and then Auxinole was replaced with Auxin. Validation of DOX and Auxin concentrations is shown in *Figure 1—figure supplement 1A and B*. The above-mentioned drug concentrations were used except where otherwise stated.

### HCT116 mAC-RIF1-L /- S
HCT116 mAC-RIF1 cells were transfected with a CRISPR/Cas9 plasmid (pX330-RIF1-29) targeting the C-terminus of exon 29 of the *RIF1* gene ('5 – CATCACCTGTTAATAAGGTAagg-3') together with a donor plasmid containing the cDNA of the C-terminal part of RIF1 (exons 30 to 35) with exon 31 (mAC-RIF1-L), or without (mAC-RIF1-S), followed by a Neomycin resistance marker (*Figure 3A*) (pmAC-RIF1-L and pmAC-RIF1-S). Transfection and clonal selection were carried out as described above. Clones were tested for RIF1 expression by western blot.

### HCT116 RIF1 KO
HCT116 cells were transfected with the same CRISPR/Cas9 plasmid that was used to construct HCT116 mAC-RIF1 cells (pX330-RIF1-N), targeting near the 1st ATG codon of the *RIF1* gene. Simultaneously transfected was a donor plasmid (pRIF1-KO) containing a hygromycin resistance marker flanked by 500 bp homology arms. Transfection and clonal selection was carried out as described above. Clones were tested for loss of RIF1 expression by western blot.

## Protein extraction and western blotting

To prepare whole cell protein extracts, cells were trypsinised and washed with Dulbecco's Phosphate-Buffered Saline (PBS) before treating with lysis buffer (10 mM Tris pH 7.5, 2 mM EDTA) containing a protease and phosphatase inhibitor cocktail (Roche). Chromatin-enriched protein fractions were prepared essentially as described (*Mailand and Diffley, 2005*). Protein concentrations were determined using the Bio-Rad RC-DC protein assay kit and equal amounts of total proteins were loaded in each lane. Proteins were transferred to PVDF membrane using the Trans-Blot Turbo Blotting System (Bio-Rad). See key resource table for antibodies used. Proteins were detected by Clarity Western ECL blotting substrate (Bio-Rad) and the Bio-Rad Chemidoc Touch Imaging System.

## Colony formation assay (CFA)

In the case of HEK293-based cell lines, cells were seeded and transfected with 50 nM RIF1 siRNA using Lipofectamine RNAimax (Invitrogen) together with Optimem (Gibco). For HCT116-based cell lines, cells were seeded with 0.2 µg/ml DOX. After two days, cells were counted using the Invitrogen Countess II FL Automated Cell Counter and 250 cells added to each well of a 6-well plate (samples always plated in technical triplicate). For HEK293-based cell lines, 1 µg/ml DOX was added to the culture medium to induce ectopic RIF1 expression whilst for HCT116-derived cell lines, 10 µM Auxin was added to the culture medium to degrade RIF1. Cells were incubated for 24 hr after which Aphidicolin (or Hydroxyurea) was added and cells incubated for a further 24 hr, before washing twice with PBS and replacement with the appropriate Aphidicolin-free medium. For HEK293-derived cell lines, 1 µg/ml supplementary DOX was re-added 72 hr after Aphidicolin removal and cells were then incubated for a further 4 days. In the case of HCT116-derived cell lines, after Aphidicolin removal, cells were incubated for 7 days in medium containing appropriate drug (DOX, Auxin, Auxinole). At the end of the incubation period, colonies consisting of more than 20 cells were counted directly using a Nikon Eclipse TS100 microscope, as in *Buonomo et al., 2009*. Values were normalised to the 0 µM APH (DMSO) control. Statistical significance was calculated using a student's T-test in Prism (Graphpad).

## Flow cytometry

To assess DNA content, cells were recovered by trypsinisation, then fixed with 70% ethanol. Cells were spun down and resuspended in 0.5 ml FxCyclePI/RNase staining solution (Molecular Probes, F10797) and incubated for 30 min at room temperature, protected from light. Manual gating on DNA content was used to distinguish cell cycle phases.

To measure the mClover fluorescence of the mAC-RIF1 fusion protein, cells were recovered by trypsinsation and fixed with a 10% neutral buffered formalin solution (Sigma, HT-5012) for 30 min at 4˚C, protected from light.

For EdU pulse labelling, 10 µM EdU was added to cells for a 15 min pulse before harvesting. The Click-iT reaction (Invitrogen) using Alexa Fluor 647 was carried out to fluorescently label EdU.

Flow cytometry was performed on a Becton Dickinson Fortessa analytical flow cytometer and FlowJo software was used for analysis. Doublet discrimination was performed by gating FSC-A against FSC-H. An equal number of events are shown in each set of histogram plots.

## Cell cycle synchronisation

HCT116 cells were seeded in 12-well dishes and treated with 20 µM Lovastatin for 24 hr to induce G1 arrest (*Rao et al., 1999*). Cells were washed and medium containing 2 mM Mevalonic acid (MVA) added to release the cells from the arrest. 8 hr after release, 9 µM RO-3306 was added to hold cells at the G2/M boundary (*Javanmoghadam-Kamrani and Keyomarsi, 2008*; *Vassilev et al., 2006*). After 28 hr in RO-3306, cells were washed and drug-free medium added allowing cells to enter mitosis. Flow cytometry was used to analyse synchronisation efficiency, and to establish and optimise the procedure for the experiment in *Figure 2* based on assessment of cell cycle progression kinetics.

## DNA fibre assay

Cells were pulse-labelled with 50 µM CldU (20 min), followed by another pulse with 250 µM IdU (20 min). After treatment with 2 mM Hydroxyurea for 4 hr, cells were harvested and lysed on a microscope slide with spreading buffer (200 mM Tris pH 7.4, 50 mM EDTA, 0.5% SDS). Slides were tilted

to allow the DNA suspension to run slowly and spread the fibres down the slide. Slides were fixed in cold (−20℃) methanol-acetic acid (3:1) and DNA denatured in 2.5 M HCl at RT for 30 min. Slides were blocked and incubated with primary antibodies for 1 hr at RT in a humidity chamber (anti-CldU, anti-IdU, anti-ssDNA). After washes with PBS, the slides were incubated with secondary antibodies (anti-rat IgG Alexa Fluor 594, anti-mouse IgG1 Alexa Fluor 488, anti-mouse IgG2a Alexa Fluor 350). Slides were air-dried and mounted with Prolong (Invitrogen). Samples were imaged under a Zeiss Axio Imager and analysed using ImageJ. CldU and IdU tract lengths were measured in double-labelled forks and the IdU/CldU ratio was used as an indicator of nascent DNA degradation. Statistical significance was calculated using a Mann-Whitney test.

## Wide-field immunofluorescence microscopy

HCT116 cells were cultured in a glass-bottomed dish (MatTek). Cells were fixed with 4% paraformaldehyde/PBS, for 15 min at room temperature. Cells were antibody stained with primary antibodies (anti-53BP1, anti-FANCD2, anti-BLM) and secondary antibodies (anti-rabbit Alexa Fluor 594, anti-goat Alexa Fluor 594, GFP-Booster ATTO 488). Microscopy was performed using a DeltaVision microscope (GE) with a x60 objective. 41 Z-stacks of 0.3 μM were taken and deconvolution was performed using the SoftWoRx software (DeltaVision). Maximum intensity projections (MIP) were created using Volocity software (PerkinElmer). Unless stated otherwise, images shown are projections.

## Live-cell imaging

HCT116 cells were cultured in a glass-bottomed dish (MatTek) containing medium without phenol red. To visualise nuclei in live cells, 0.5 μM SiR-DNA (Spirochrome) was added to the medium before observation. Live-cell imaging was performed using a DeltaVision microscope equipped with an incubation chamber and a $CO_2$ supply (GE). Image analysis and quantification was performed using the Volocity software (PerkinElmer).

## Confocal immunofluorescence microscopy

HCT116 cells and U2OS cells were cultured in ibiTreat μ-slide eight well dishes (Ibidi). Cells were treated with 1 μM Aphidicolin for 24 hr after which Aphidicolin was removed and cells were incubated for a further 12 hr. Where indicated siTopBP1 (Ambion s21823, see key resource table for sequence) transfection was performed 48 hr before Aphidicolin treatment. Cells were fixed with either a 10% neutral buffered formalin solution (Sigma, HT-5012) for 10 min at room temperature or with 100% methanol for 15 min at −20℃. Blocking was for 30 min with PBST/1%BSA at 4℃ after which antibody staining was performed (anti-cyclinA2-Alexa Fluor 555, anti-53BP1 self-conjugated to Alexa Fluor 647). Slides were mounted with Ibidi mounting medium containing DAPI. Confocal microscopy was performed using a LSM880 + Airyscan (Zeiss) with a x63 objective. Z-stacks were imaged (40 slices). Images were processed first to an Airyscan image and then to a MIP using ZEN Black (Zeiss). Cyclin A2 staining was used to identify G1 cells (Cyclin A2 negative = G1 cells). Both image analysis (colocalisation) and quantification were performed using CellProfiler (Broad Institute) with a custom-made pipeline. The 'IdentifyPrimaryObjects' module was used to identify individual cells (DAPI positive), the G1 cell population (CyclinA2 negative), mAC-RIF1 foci (mClover positive) and 53BP1 foci (Alexa Fluor 647 positive). The 'RelateObjects' module was used to count the number of mAC-RIF1 and 53BP1 foci in a cell and to assess colocalisation of 53BP1 foci with mAC-RIF1 foci. Statistical significance was calculated using a one-way ANOVA using Prism (Graphpad).

## cDNA analysis of RIF1 splice variants

A human cDNA panel covering 48 major tissues was obtained from Insight Biotechnology (Origene HMRT104). The two RIF1 splicing variants were amplified by competitive PCR using a single primer pair (see key resource table). The PCR products were analysed on a 2% agarose gel (3:1 mix of low-melting temperature agarose and normal agarose). Bands were visualised by staining with SYBR Green I dye, and fluorescent images were captured using Bio-Rad ChemiDoc Touch system. Intensities of bands specific to RIF1-L, RIF1-S and an additional minor band caused by heteroduplex formation, were measured using ImageLab software (Bio-Rad). The relative abundance of RIF-L and RIF1-S were calculated based on the band intensities, with 50% of the heteroduplex band intensity assigned to RIF1-L and 50% assigned to RIF1-S.

## Mitotic enrichment for mitotic bridge analysis

HCT116 cells were treated with 0.1 µM ICRF-193 (Topoisomerase II inhibitor) (Sigma) for 30 min. Alternatively cells were treated with 1 µM Aphidicolin for 24 hr after which Aphidicolin was removed and cells were incubated for a further 12 hr. Mitotic shake-off was performed to release mitotic cells, which were then spun down onto a slide. Cells were fixed with 10% neutral buffered formalin solution (Sigman, HT-5012) for 10 min at room temperature. Blocking was for 30 min with 5% FBS/PBS followed by primary antibody staining (anti-PICH) and secondary antibody staining (AF647). Slides were mounted with Prolong Glass mounting medium (Invitrogen). Z-stack (10 optical sections) images were acquired using an LSM880 + Airyscan (Zeiss) with a x63 objective, and processed as above. UFBs were counted manually and defined as PICH-positive, DNA-negative threads between separating chromatin masses. Chromosome bridges were counted manually and defined as DNA-positive threads between separating chromatin masses.

## Preparing metaphase spreads

HCT116 cells were treated with 0.4 µM Aphidicolin for 24 hr after which 0.1 µg/ml colcemid (Gibco) was added in parallel with 40 µM EdU for 30 min before harvesting. Chromosome preparation was performed by incubating cells with 75 mM KCl for 10 min followed by fixing cells three times with methanol:glacial acetic acid (3:1). The chromosome preparation was dropped onto slides and the Click-iT reaction (Invitrogen) using Alexa Fluor 488 was performed to fluorescently label EdU. Slides were mounted with Vectashield and imaged using a Zeiss Epiflourescence microscope using a x100 objective. Images were manually examined for apparent breaks or gaps as well as EdU incorporation in mitotic chromosomes, and 'per chromosome' values were calculated. Statistical significance was calculated using a Mann-Whitney test (Prism Graphpad).

## Contact for reagent sharing

Request for reagents or further information should be directed to and will be fulfilled by the Lead Contact, Anne Donaldson (a.d.donaldson.ac.uk).

## Acknowledgements

Thanks to members of the Aberdeen Chromosome Biology Group for helpful comments, and Ronan Broderick and Wojciech Niedzwiedz for advice on mitotic bridge analysis. We thank Raif Yuecel and his team at the Iain Fraser Cytometry Centre for assistance, and Kevin Mackenzie and his team at the Microscopy and Histology Core Facility. Work was supported by Cancer Research UK Studentship Award C1445/A20596 and CRUK Programme Award C1445/A19059; by JSPS KAKENHI Grants Numbers 17K15068, 18H02170 and 18H04719; by research grants from the Daiichi Sankyo's Foundation of Life Science and the Takeda Science Foundation; and by the UK Medical Research Council (MC_UU_00007/13). Collaboration was supported by a 2017 JSPS Summer Programme Fellowship.

## Additional information

### Funding

| Funder | Grant reference number | Author |
| --- | --- | --- |
| Cancer Research UK | C1445/A20596 | Anne D Donaldson |
| Cancer Research UK | C1445/A19059 | Anne D Donaldson |
| Japan Society for the Promotion of Science | 17K15068 | Masato T Kanemaki |
| Japan Society for the Promotion of Science | 18H02170 | Masato T Kanemaki |
| Japan Society for the Promotion of Science | 18H04719 | Masato T Kanemaki |
| Medical Research Council | MC_UU_00007/13 | Nick Gilbert |
| Daiichi Sankyo Foundation of | Research | Masato T Kanemaki |

Life Science

| Takeda Science Foundation | Research | Masato T Kanemaki |
| Japan Society for the Promotion of Science | Summer Programme Fellowship | Lotte P Watts |

The funders had no role in study design, data collection and interpretation, or the decision to submit the work for publication.

## Author contributions

Lotte P Watts, Conceptualization, Funding acquisition, Validation, Investigation, Visualization, Methodology, Writing - original draft, Project administration, Writing - review and editing; Toyoaki Natsume, Conceptualization, Investigation, Visualization, Methodology; Yuichiro Saito, Lora Boteva, Investigation, Methodology; Javier Garzon, Investigation, Visualization, Methodology; Qianqian Dong, Validation; Nick Gilbert, Conceptualization, Funding acquisition; Masato T Kanemaki, Conceptualization, Supervision, Funding acquisition, Investigation, Methodology; Shin-ichiro Hiraga, Conceptualization, Supervision, Funding acquisition, Investigation, Visualization, Methodology, Writing - original draft, Project administration, Writing - review and editing; Anne D Donaldson, Conceptualization, Supervision, Funding acquisition, Writing - original draft, Writing - review and editing

## Author ORCIDs

Lotte P Watts https://orcid.org/0000-0002-1364-4345
Toyoaki Natsume http://orcid.org/0000-0002-3544-4491
Masato T Kanemaki http://orcid.org/0000-0002-7657-1649
Shin-ichiro Hiraga https://orcid.org/0000-0002-8722-3869
Anne D Donaldson https://orcid.org/0000-0001-7842-8136

## Decision letter and Author response

Decision letter https://doi.org/10.7554/eLife.58020.sa1
Author response https://doi.org/10.7554/eLife.58020.sa2

# Additional files

## Supplementary files

• Source data 1. Underlying data for graphs.

• Transparent reporting form

## Data availability

All data generated or analysed during this study are included in the manuscript and supporting files.

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
