## [Decision Letter]

**Acceptance summary:**

It is becoming very clear that the conserved Rif1 protein is increasingly important in mechanisms that all appear to be geared towards maintaining genome stability. The specific mechanisms include DNA replication initiation, replication fork stability or drug sensitivity, and some are associated with DSB end protection and DNA repair pathway choice. In this manuscript, the Donaldson lab presents exciting and completely unanticipated evidence for splice-isoform specific functions of Rif1 in human cells. A shorter isoform (Rif1S) appears to be able to fulfill the replication initiation and fork associated functions, but only the longer Rif1L also is able to protect chromosomes from potential damage incurred by DNA replication inhibitors (Aphidicolin or HU). This function appears required during S-phase but also after S in the formation of the 53BP1-nuclear bodies in the next G1. These results thus are remarkable for both the DNA replication and repair fields as they possibly open up a new way of thinking about how Rif1 functions extend to the next G1 phase and the 53BP1 NB formation.

**Decision letter after peer review:**

Thank you for submitting your article "The RIF1-Long splice variant promotes G1 phase 53BP1 nuclear bodies to protect against replication stress" for consideration by *eLife*. Your article has been reviewed by three peer reviewers, one of whom is a member of our Board of Reviewing Editors, and the evaluation has been overseen by Kevin Struhl as the Senior Editor. The reviewers have opted to remain anonymous.

The reviewers have discussed the reviews with one another and the Reviewing Editor has drafted this decision to help you prepare a revised submission.

The highly conserved Rif1 protein functions in an increasing variety of mechanisms that all appear to be geared towards guarding genome stability. Some of them affect DNA replication initiation, fork stability or DNA replication drug sensitivity, while others are associated with DSB end protection and DNA repair pathway choice. However, molecular mechanisms describing how Rif1 works in these seemingly disparate mechanisms are relatively sparse. In this manuscript, the Donaldson lab present exciting evidence for a splice-isoform specific function in human cells. Indeed, while a slightly shorter isoform (Rif1S, lacks exon 31, 2446 a.a.) appears to be able to fulfill the replication initiation and replication fork associated functions, only the Rif1L form (includes exon 31, 2472 a.a.) also is able to protect chromosomes from potential damage incurred by DNA replication inhibitors (Aphidicolin or HU). This function appears required during S-phase but also after S in the formation of the 53BP1-nuclear bodies in the next G1. Therefore, these experiments document isoform specific functions for human Rif1 that were completely unanticipated.

All reviewers appreciated the exciting, strong and conclusive evidence around the isoform specific functions for the Rif1L versus the Rif1S. They also appreciate the novelty of these findings for the DNA replication and repair fields as they possibly open up a new way of thinking about how Rif1 functions extend to the next G1 phase and the 53BP1 NB formation. They therefore unanimously thought that the paper could develop into a strong candidate for *eLife*. However, we strive to publish the most influential and solid papers and in that line of thinking, certain aspects of the work must be strengthened. After a discussion about how to solidly bring this paper up to the next level, the reviewers request:

1) The link between the isoform specific function of Rif1L and the 53BP1 NB in G1 was perceived as most exciting, but also needing some more substantiating. In order to support/test the claims of paucity of 53BP1-NBs around under-replicated DNA in RIF1-L deficient cells, you should provide additional molecular connections, such as processes important at fragile sites. For example, you could monitor UFB formation, FANCD2 twin foci on UFBs, MiDAS.

2) Data on global DNA synthesis, fork speed, fork density in RIF1-S or RIF1-L expressing cells is missing. This could be derived with appropriate DNA combing experiments and reviewers thought that some of those data may already be in your hands.

3) A big question is the molecular mechanism by which the 26aa region unique to the Rif1L form functions. While fully determining the mechanism is perhaps beyond the scope of this format, the reader is left hanging. Is the sequence interesting, similar to anything else, found in protein-protein interaction screens, or are there known modifications at any of those positions?

4) Most of the data are presented in bar charts which is not the best method. A modern data presentation that reveals the distribution and reproducibility across independent replicates would be much better. The number of independent biological (not technical) replicates is not reported. A recent discussion of the shortcomings of bar graphs (Lord et al. JCB 2020 https://doi.org/10.1083/jcb.202001064)

Just to be clear, the reviewers do expect you to fully address these above points, even with new experimentation, as indicated.

Reviewers made additional comments on certain aspects of the work which you will find compiled below. Those points are for your perusal, to be incorporated at least textually if possible, but are deemed not essential to be addressed by experiments.

5) Do RIF1-L deficient cells show other upstream regulators of 53BP1-NBs such as pATM, RNF168? You could also check for the presence of γH2AX in G1 (another proxy for 53BP1-NBs) and extend this analysis to S and G2 phase (as a proxy for unrepaired DNA lesions) to get an overview to understand the consequence of defective 53BP1-NBs around under-replicated DNA.

6) Authors should monitor the total as well nuclear levels of 53BP1 protein in RIF1-L deficient cells.

7) You could provide more evidence to link the defective 53BP1-NBs formation and aphidicolin sensitivity. A recent study has suggested that RIF1 loss leads to unprotected replication factories (Ribeyre et al., 2020). You could also check whether such local alteration in replication factories is evident in RIF1-L deficient cells. (Ref.: Ribeyre, C., Lebdy, R., Patouillard, J., Larroque, M., Abou-Merhi, R., Larroque, C., and Constantinou, A. (2020). The isolation of proteins on nascent DNA (iPOND) reveals a role for RIF1 in the organization of replication factories. bioRxiv, 669234.)

8) The function unique to Rif1L (vs. Rif1S) is confined to two phenotypes: colony forming ability after replication stress and 53BP1 nuclear foci. I appreciate that the authors have done considerable work to show how the two forms are similar (origin firing, MCM dephosphorylation, etc.). More characterization of this phosphatase-independent function through additional stainings or assays would greatly strengthen the paper.

9) In Figure 2C, FACS analyses are shown for cell cycle distribution of cells in the experiments. Given that the execution point approaches here do rely on robust and quantitative cell cycle arrest phenotypes, the FACS profiles are of limited use. I would prefer them being shown as % cells in the respective cell cycle phases (i.e. % G1, % S, % G2) for each experimental time point.

10) Is there a Rif1 immunoblot available for the experiment in Figure 3E?

11) The tubulin blot scan in Figure 1E appears to have lost resolution in processing (very pixelated).

---

## [Author Response]

All reviewers appreciated the exciting, strong and conclusive evidence around the isoform specific functions for the Rif1L versus the Rif1S. They also appreciate the novelty of these findings for the DNA replication and repair fields as they possibly open up a new way of thinking about how Rif1 functions extend to the next G1 phase and the 53BP1 NB formation. They therefore unanimously thought that the paper could develop into a strong candidate for eLife. However, we strive to publish the most influential and solid papers and in that line of thinking, certain aspects of the work must be strengthened. After a discussion about how to solidly bring this paper up to the next level, the reviewers request:1) The link between the isoform specific function of Rif1L and the 53BP1 NB in G1 was perceived as most exciting, but also needing some more substantiating. In order to support/test the claims of paucity of 53BP1-NBs around under-replicated DNA in RIF1-L deficient cells, you should provide additional molecular connections, such as processes important at fragile sites. For example, you could monitor UFB formation, FANCD2 twin foci on UFBs, MiDAS.

We have now expanded considerably on the initial observations (from Supplementary Figure 3A of the original manuscript), by adding two new Figures which address this comment: revised Figure 4 which examines UFBs and chromosome bridges after drug treatment, and revised Figure 4—figure supplement 1 which examines chromatin breaks and MiDAS synthesis.

Revised Figure 4 shows that in our HCT116-based isoform-specific cell lines, ICRF-193 treatment induces mainly UFBs, and Aphidicolin treatment mainly chromosome bridges (that stain with DAPI). Their rates of formation are not affected by the RIF1 isoform present, and both RIF1-L and RIF1-S localise to the UFB and chromosome bridge structures induced.(Our finding that ICRF-193 is more effective than Aphidicolin in inducing UFBs is consistent with previous studies, which mainly use ICRF-193 for UFB induction e.g. Nielsen et al. 2015 Nature Comm 6: 8962; Hengeveld et al., 2015)

Figure 4—figure supplement 1 shows that complete deletion of RIF1 has no significant effect on MiDAS or on the incidence of chromatin breaks, making it very unlikely that the RIF1 isoforms would have differential effects on MiDAS or chromatin breakage.

These new figures are discussed in the subsection “RIF1-Long and -Short isoforms do not appear to differ in mitotic functions”. Overall, these new analyses do not suggest that the differential capability of RIF1 isoforms in protecting against Aphidicolin is caused by a differential role of the isoforms affecting mitotic bridges, chromatin breaks, or MiDAS sites. Based on the results shown in revised Figures 4 and 5 and the supplements to these figures, we suspect that the defect in RIF1-L-deficient cells involved failure to recognise damage sites that *remain* problematic *after* these mitotic processes, and to handle them correctly as cells progress into and through the next cell cycle—in a process involving 53BP NBs and possibly TopBP1 (please see also new data presented in response to point 5 below).

2) Data on global DNA synthesis, fork speed, fork density in RIF1-S or RIF1-L expressing cells is missing. This could be derived with appropriate DNA combing experiments and reviewers thought that some of those data may already be in your hands.

In revised Figure 3—figure supplement 4, we now include flow cytometry data showing global DNA synthesis rate (part C), fork speed data (B), and measurement of PCNA loading on chromatin (reflective of fork density) (D). The experiments are discussed in the fourth paragraph of the subsection “Only the RIF1-Long splice isoform protects cells from replication stress”.

3) A big question is the molecular mechanism by which the 26aa region unique to the Rif1L form functions. While fully determining the mechanism is perhaps beyond the scope of this format, the reader is left hanging. Is the sequence interesting, similar to anything else, found in protein-protein interaction screens, or are there known modifications at any of those positions?

We now discuss this issue in the fifth paragraph of the Discussion. The 26 aa region unique to RIF1-L contains a ‘phosphoSPxF’ BRCT interaction motif, raising the possibility that RIF1-L interacts with one of the several BRCT domain-containing proteins involved in chromosome maintenance. However at present we do not know which BRCT protein would be the relevant target, since initial IP experiments did not suggest a likely candidate.

4) Most of the data are presented in bar charts which is not the best method. A modern data presentation that reveals the distribution and reproducibility across independent replicates would be much better. The number of independent biological (not technical) replicates is not reported. A recent discussion of the shortcomings of bar graphs (Lord et al. JCB 2020 https://doi.org/10.1083/jcb.202001064)

We now present most of the relevant experimental data (mainly CFA experiments) in line graph form showing reproducibility across replicates, and give numbers of biological/technical replicates for each experiment in the figure legends. In most cases we also show the separate biological replicate experiments in the supplementary figures. We have retained the ‘bar graph’ format in Figure 2D and Figure 5B and C where it is helpful for interpretation, but in these cases the plots also now show the distribution and reproducibility across replicates.

Just to be clear, the reviewers do expect you to fully address these above points, even with new experimentation, as indicated.Reviewers made additional comments on certain aspects of the work which you will find compiled below. Those points are for your perusal, to be incorporated at least textually if possible, but are deemed not essential to be addressed by experiments.5) Do RIF1-L deficient cells show other upstream regulators of 53BP1-NBs such as pATM, RNF168? You could also check for the presence of γH2AX in G1 (another proxy for 53BP1-NBs) and extend this analysis to S and G2 phase (as a proxy for unrepaired DNA lesions) to get an overview to understand the consequence of defective 53BP1-NBs around under-replicated DNA.

A preliminary examination of phospho-ATM and RNF8, known upstream regulators of 53BP1 nuclear bodies, revealed no evident differences in their behaviour in cells containing RIF1-L and RIF1-S. TopBP1 has also been suggested as involved in correct recognition of damage sites and their transfer into 53BP1 NBs (Pedersen et al., 2015), so we tested for any effect of TopBP1 on 53BP1 bodies by RIF1, and found that TopBP1 depletion causes a defect in RIF1-L localisation to 53BP1 NBs (presented in revised Figure 5—figure supplement 2). This observation may suggest how damage is recognised and carried through by RIF1-L to 53BP1 NBs. Based on IP studies we have not however found evidence for a direct interaction of RIF1-L with TopBP1, suggesting the effect could be indirect and making further elaboration on the connection between TopBP1 and RIF1-L function beyond the scope of the present study.

6) Authors should monitor the total as well nuclear levels of 53BP1 protein in RIF1-L deficient cells.

We now include a western blot showing total levels of 53BP1 in the various cell lines (Figure 5—figure supplement 1A).

7) You could provide more evidence to link the defective 53BP1-NBs formation and aphidicolin sensitivity. A recent study has suggested that RIF1 loss leads to unprotected replication factories (Ribeyre et al., 2020). You could also check whether such local alteration in replication factories is evident in RIF1-L deficient cells. (Ref.: Ribeyre, C., Lebdy, R., Patouillard, J., Larroque, M., Abou-Merhi, R., Larroque, C., and Constantinou, A. (2020). The isolation of proteins on nascent DNA (iPOND) reveals a role for RIF1 in the organization of replication factories. bioRxiv, 669234.)

We now supply data measuring several additional S phase parameters as explained in our response to point 2, in which we find no difference in the effect of RIF1-L and RIF1-S in effects on replication events. IPOND analysis of replication factories would involve a major new set of experiments not feasible in the timescale of this revision (and actually it is not clear to us how directly such experiments might strengthen the link to defective 53BP1 NB formation).

8) The function unique to Rif1L (vs. Rif1S) is confined to two phenotypes: colony forming ability after replication stress and 53BP1 nuclear foci. I appreciate that the authors have done considerable work to show how the two forms are similar (origin firing, MCM dephosphorylation, etc.). More characterization of this phosphatase-independent function through additional stainings or assays would greatly strengthen the paper.

Please see the additional assays and measurements provided and mentioned in response to comments 1, 2, and 5.

9) In Figure 2C, FACS analyses are shown for cell cycle distribution of cells in the experiments. Given that the execution point approaches here do rely on robust and quantitative cell cycle arrest phenotypes, the FACS profiles are of limited use. I would prefer them being shown as % cells in the respective cell cycle phases (i.e. % G1, % S, % G2) for each experimental time point.

We now provide tables showing the % of cells in each phase alongside the flow cytometry data, in revised Figure 2C and revised Figure 2—figure supplement 2B.

10) Is there a Rif1 immunoblot available for the experiment in Figure 3E?

We do not have a RIF1 blot for this particular experiment, but the efficacy of both the HCT116 and HEK293 systems is well-established, as shown in revised Figure 3B, Figure 3—figure supplement 3B, and in Hiraga et al., 2017.

11) The tubulin blot scan in Figure 1E appears to have lost resolution in processing (very pixelated).

This defect has been corrected.